# LLM-generated messages can persuade humans on policy issues

Hui Bai [1,2] ✉, Jan G. Voelkel [1,3,4], Shane Muldowney [1],
Johannes C. Eichstaedt[5] & Robb Willer [1,3] ✉

The emergence of large language models (LLMs) has made it possible for generative artificial intelligence (AI) to tackle many higher-order cognitive tasks, with critical implications for industry, government, and labor markets. Here, we investigate whether existing, openly-available LLMs can be used to create messages capable of influencing humans' political attitudes. Across three pre-registered experiments (total $N = 4829$), participants who read persuasive messages generated by LLMs showed significantly more attitude change across a range of policies - including polarized policies, like an assault weapons ban, a carbon tax, and a paid parental-leave program - relative to control condition participants who read a neutral message. Overall, LLM-generated messages were similarly effective in influencing policy attitudes as messages crafted by lay humans. Participants' reported perceptions of the authors of the persuasive messages suggest these effects occurred through somewhat distinct causal pathways. While the persuasiveness of LLM-generated messages was associated with perceptions that the author used more facts, evidence, logical reasoning, and a dispassionate voice, the persuasiveness of human-generated messages was associated with perceptions of the author as unique and original. These results demonstrate that recent developments in AI make it possible to create politically persuasive messages quickly, cheaply, and at massive scale.

Recent developments in generative artificial intelligence (AI)—in particular, large language models (LLMs)—have led to major breakthroughs, achieving capacities thought to be impossible just a few years ago. AI-driven applications can now be used to create visual art[1], compose music[2], write computer source code[3], and produce text of striking complexity[4]. LLMs, specifically, have proven capable of debating with humans[5] and outperforming humans in online strategy games involving negotiation[6], suggesting that the most advanced language models can now generate complex reasoning and language expression at human or near-human levels.

With LLMs reaching these and other higher-order capacities, their potential influence on politics has emerged as a high-stakes question demanding scholarly attention. AI-generated images have already been deployed in political advertisements in the US[7], indicating that LLM-generated text is likely to appear in political campaigns soon, if not already.

While there are many potential applications of LLMs to political persuasion, it is unknown whether LLM-generated content can be used to shift humans' views on political issues. Understanding the capacity of LLMs for political persuasion is crucial to inform the urgency with which lawmakers may need to develop regulations, and LLM developers may need to implement guardrails to encourage benevolent applications and constrain worrisome applications.

[1]Politics and Social Change Lab, Stanford University, Stanford, CA, USA. [2]Political Belief Lab, Minnetonka, MN, USA. [3]Department of Sociology, Stanford University, Stanford, CA, USA. [4]Brooks School of Public Policy, Cornell University, Ithaca, NY, USA. [5]Department of Psychology & Institute for Human-Centered AI, Stanford University, Stanford, CA, USA. ✉e-mail: maxhuibai@politicalbelieflab.com; willer@stanford.edu

Here, we empirically examine whether publicly available LLMs can generate messages supporting policies that can persuade humans to shift their views on these policies. We had competing expectations for whether LLM-generated messages would be persuasive in this way, and also how their persuasiveness would compare to that of messages written by lay humans. While the capacities of LLMs have advanced tremendously in recent years, there are also reasons to doubt that exposure to LLM-generated messages would shift people's political views. The best available experiments on political persuasion generally find that human-authored messages have small effects[8,9], and the effects of persuasive efforts by political campaigns are typically small or null[10]. Further, political persuasion is a complex activity, potentially drawing upon a range of advanced skills, including perspective-taking, knowledge of the topic, logical reasoning, clarity of expression, and knowledge of effective interpersonal influence techniques, with success ultimately determined by the response of the person receiving the persuasive appeal. Persuasion is also uniquely challenging in highly polarized settings, such as the contemporary US, where many views are strongly held and difficult to influence[11].

It is also possible, however, that LLM-generated messages are as persuasive as, or perhaps even more persuasive than, messages written by lay humans: LLMs can be used to produce language with a proficiency often surpassing average human abilities in generating complex, coherent, and topical text[3]. While human-authored messages generally show small effect sizes in political persuasion experiments, LLMs may exceed humans. LLMs' data-driven approach, derived from vast corpora of written language, may enable these models to dispassionately identify the strongest evidence and reasoning from their knowledge base and implement the most effective persuasion strategies more systematically than humans. Because of these competing expectations on whether LLM-generated messages can persuade humans, we did not specify an a priori hypothesis about whether messages generated by LLMs might be more or less persuasive than those generated by humans.

Despite much speculation regarding future applications of AI to politics, research has not yet tested whether exposure to LLM-generated messages can shift people's views on policy issues. On the topic of AI-human interaction, several lines of prior work addressed related, but qualitatively distinct, themes. First, researchers have investigated if text attributed to AI versus humans is evaluated differently on attributes such as perceived credibility[12,13]. In these studies, researchers investigate the role of perceived authorship (AI versus human) as opposed to the effects of the texts. In more recent research, researchers have also begun to investigate how people rate messages generated by LLMs for their perceived persuasiveness in advertisements[14] or for promoting vaccine adoption (for example, as measured by agreement with statements like, "The message gave a reason for getting a COVID vaccine that is important to me"[15]). Second, researchers have explored the role of automated systems, predominantly Twitter bots, in propagating human-authored misinformation on social media[16,17]. Third, researchers have found that AI-generated autocomplete suggestions can shift humans' writing, and the writers' own private opinions, across a range of political and societal topics, often outside the human writer's awareness[18,19]. Broadly speaking, these studies demonstrated that automated systems can disseminate human-created misinformation (e.g., by retweeting), but none show that LLMs can be used to create original content that influences political opinions (see Supplementary Table 12 for a summary of related research).

These distinctions are not trivial, practically or scientifically. Practically, the most widely discussed applications of AI-generated persuasive content—for example, AI-generated content being used by political campaigns—would involve messages intended to shift attitudes on political issues and topics. Testing existing LLMs' capability to generate messages that can shift attitudes is a more direct evaluation of such applications. Theoretically, prior research finds positive views of messages and the ability of messages to persuade people are often positively associated, but increasing positive views of messages does not necessarily result in attitude change[20,21]. Thus, for our goal of testing the capacity to persuade using AI-generated messages, prior research has not yet conclusively demonstrated persuasion[22].

The current project focuses on persuasion in support of a range of proposed policies. Establishing the capacity for LLM-generated messages to persuade people on policy issues, particularly polarized policies, is societally significant. Political persuasion is a phenomenon of interest in academia and beyond. It is a major area of academic study in the fields of social psychology and political science, a multibillion-dollar global industry, and an important path to a number of societally impactful behaviors, including voting for candidates as well as on ballot initiatives, and social movement participation[11]. Here, we examine whether LLM-generated political appeals can be used to persuade humans to change their attitudes on policy issues.

Across three pre-registered survey experiments (total $N = 4829$) conducted in November and December 2022 on diverse, national samples of Americans, including one (Study 3) that was representative of the US population across a range of demographic benchmarks (see "Methods"), we tested whether LLM-generated political messages can be used to persuade humans to shift their attitudes towards various policies. Participants in Studies 1 and 2 were randomly assigned to either read a persuasive message on a policy generated by the LLM GPT-3 and 3.5[23] (LLM condition), a persuasive message written by a lay human participant (Human condition), a message chosen by a prior human participant from a set of five LLM-generated messages (Human-in-the-Loop condition), or a neutral message on an irrelevant topic (e.g., about the history of skiing; Control condition). Study 3 included only an LLM condition and a Control condition (see "Covariate Balance Checks" in Supplementary Methods for covariate balance checks for each study).

To increase generalizability, we studied responses to messages across a range of policy issues[24]: a public smoking ban in Study 1, an assault weapons ban in Study 2, and, in Study 3, one of four randomly-assigned policies—a carbon tax, an increase to the child tax credit, a paid parental leave program, and automatic voter registration. The policies were briefly explained to participants before they reported their baseline support for the policies. These policies covered a range of topics, including economic and social policies, and policies for which there is high versus low partisan polarization in the American public (see "Distribution of Pre-Treatment Levels of Support for Policies" in Supplementary Methods). In all experiments, participants reported their support for a policy before and after reading the assigned message. We pre-registered our research questions and analysis strategy for all three experiments.

## Results

We regressed participants' post-treatment policy support on experimental condition (coded as a series (Studies 1 and 2) or a single (Study 3) dummy variable using the Control condition as the reference category). We controlled for pre-treatment policy support in all studies and for message topic in Study 3 (the only study in which we studied multiple topics). All statistical tests are two-tailed. Detailed descriptions of the statistical tests can be found in the "Methods" section and full results can be found in the Supplementary Information.

### Policy support

Across all three studies, exposure to LLM-generated messages consistently led to attitudinal change among human participants. As is common in the political persuasion literature[8-11], as well as research in the psychology of language[25,26], effect sizes were consistently small, ranging from about 2 to 4 points on the 101-point policy support scales we used in the three experiments (see Fig. 1; also see Supplementary

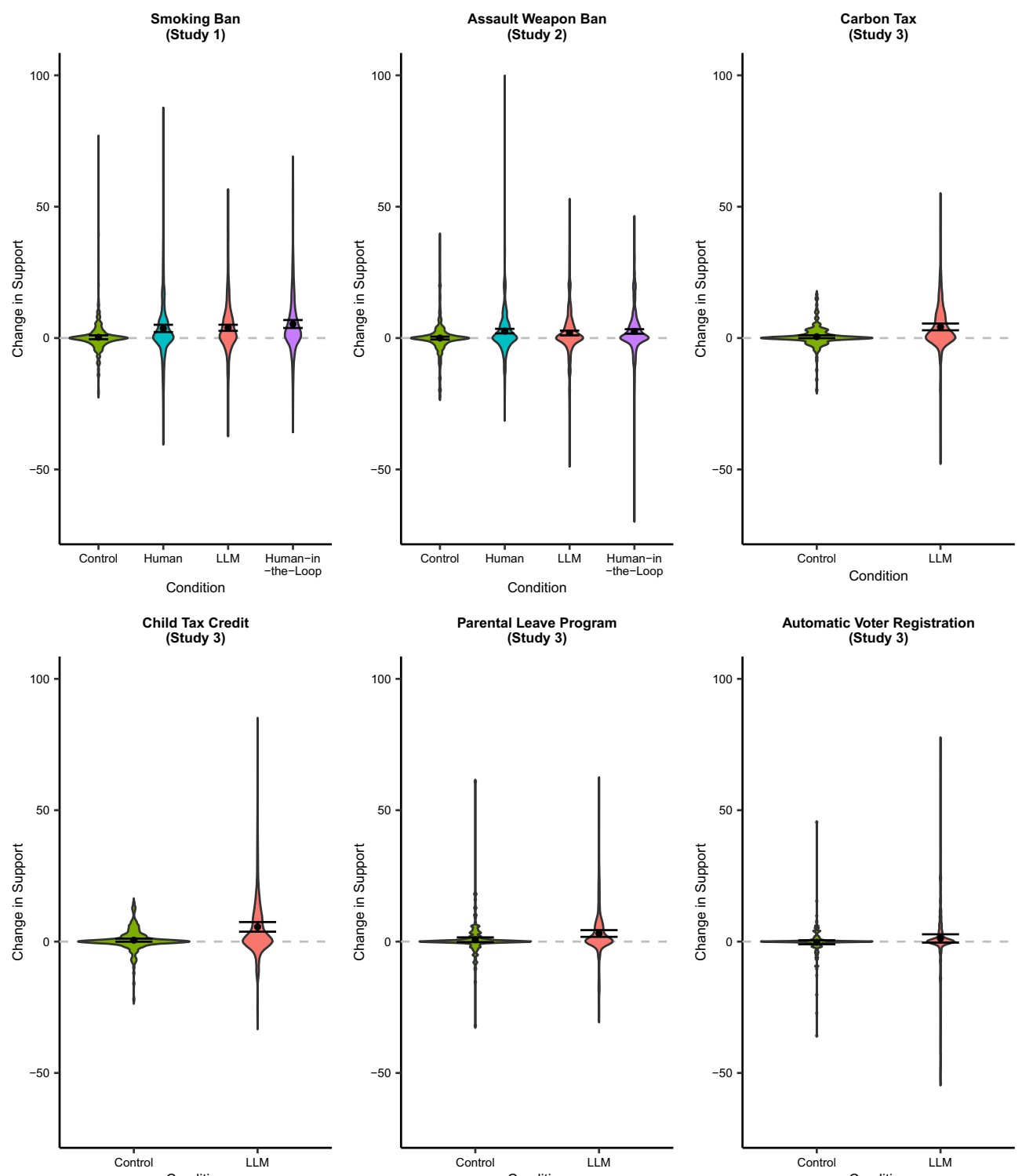

**Fig. 1 | Participants' change in policy support by condition across studies.** Y-axes represent the average difference between participants' post-treatment and pre-treatment policy support (both scaled from 0 to 100, 100 = highest level of support). Higher scores indicate participants became more supportive of the policy. Data are analyzed using regression, two-tailed, and results are presented as mean values with 95% confidence intervals. $N_{Study\ 1}$ = 1203 participants, $N_{Study\ 2}$ = 2016 participants, $N_{Study\ 3}$ = 1610 participants.

Fig. 3 for the distribution of the messages' persuasiveness which shows the difference between participants' post-treatment and pre-treatment policy support across the Human and LLM conditions in Studies 1 and 2). In Study 1, participants supported a smoking ban significantly more if they were assigned to the LLM condition than if they were assigned to the Control condition ($t(1198)$ = 4.17, $p$ < 0.001, $b$ = 3.62, 95% CI =

[1.92, 5.32]; see Supplementary Table 1a). Study 2 replicated this effect using the highly polarized topic of gun control. Participants supported an assault weapons ban significantly more if they were assigned to the LLM condition than if they were assigned to the Control condition ($t(2011)$ = 3.16, $p$ = 0.002, $b$ = 1.81, 95% CI = [0.69, 2.93]; see Supplementary Table 2a). Study 3 showed the robustness of this effect across

a number of polarizing issues ($t(1604) = 7.35$, $p < 0.001$, $b = 3.15$, 95% CI = [2.31, 3.99]; see Supplementary Table 3a for the overall model as well as the issue-specific results).

Additionally, exposure to LLM-generated messages was as influential on participants' policy views as exposure to human-generated messages. Participants in the Human condition also supported a smoking ban and gun control significantly more than participants in the Control condition (Study 1: $t(1198) = 3.90$, $p < 0.001$, $b = 3.36$, 95% CI = [1.67, 5.05]; Study 2: $t(2011) = 4.13$, $p < 0.001$, $b = 2.35$, 95% CI = [1.23, 3.47]; see Supplementary Tables 1a and 2a). The effects of exposure to LLM-generated messages and human-generated messages on policy support were similar in magnitude (Study 1: $t(892) = 0.27$, $p = 0.787$, $b = 0.26$, 95% CI = [−1.60, 2.12], Bayes Factor (BF01) = 24.57; Study 2: $t(1501) = −0.84$, $p = 0.403$, $b = −0.52$, 95% CI = [−1.74, 0.70], BF01 = 22.74; see Supplementary Tables 1a and 2a).

Participants assigned to read one of the LLM-generated messages selected by human participants in the Human-in-the-Loop condition also supported a smoking ban and gun control more than participants in the Control (Study 1: $t(1198) = 5.56$, $p < 0.001$, $b = 5.04$, 95% CI = [3.26, 6.82]; Study 2: $t(2011) = 4.12$, $p < 0.001$, $b = 2.35$, 95% CI = [1.23, 3.46]; see Supplementary Tables 1a and 2a). We also tested for a possible difference in the persuasiveness of the Human-in-the-Loop messages compared to the messages participants received in the LLM condition, in both a meta-analysis ($t(1571) = 1.94$, $p = 0.052$, $b = 0.94$, 95% CI = [−0.01, 1.88]), as well as in the individual studies (Study 1: $t(573) = 1.51$, $p = 0.131$, $b = 1.45$, 95% CI = [−0.43, 3.34], BF01 = 7.61; Study 2: $t(996) = 0.84$ $p = 0.403$, $b = 0.52$, 95% CI = [−0.70, 1.74], BF01 = 22.79; see Supplementary Tables 1a and 2a). Participants in the Human-in-the-Loop condition did not significantly differ in support for the two policies from participants in the Human condition (Study 1: $t(892) = 1.70$, $p = 0.089$, $b = 1.68$, 95% CI = [−0.26, 3.62], BF01 = 6.44; Study 2: $t(1501) = 0.02$, $p = 0.984$, $b = 0.01$, 95% CI = [−1.20, 1.23], BF01 = 31.82; see Supplementary Tables 1a and 2a; meta-analysis: $t(1588) = 0.73$, $p = 0.466$, $b = 0.56$, 95% CI = [−0.94, 2.06].)

## Moderating variables

We also found evidence in exploratory analyses that the effects of LLM-generated messages were moderated by participants' partisan identity. Aggregating data from Studies 1 and 2, we found that partisan identity (measured as a 7-point variable, re-coded to range from 0 = Strong Democrat to 1 = Strong Republican) moderated the treatment effects. The significant Partisan Identity × LLM (vs Control) condition interaction effect ($t(3209) = −3.37$, $p < 0.001$, $b = −5.16$, 95% CI = [−8.16, −2.16]) suggests that participants were more persuaded by the LLM-generated messages if they were more strongly identified as Democrat. We found a significant interaction effect for the Human condition as well, indicating a similar pattern in which the messages were more persuasive if the participants were more strongly identified as Democrats: Partisan Identity × Human (vs Control) condition ($t(3209) = −2.55$, $p = 0.011$, $b = −3.93$, 95% CI = [−6.95, −0.90]). The Human-in-the-Loop condition × Partisan Identity interaction effect was not significant ($t(3209) = −1.55$, $p = 0.122$, $b = −2.42$, 95% CI = [−5.48, 0.65]).

In other moderation analyses, we also found evidence that pre-treatment policy support moderated the effect of LLM-generated messages. The pre-treatment policy support × LLM condition interaction effect was positive and significant ($t(3209) = 3.21$, $p = 0.001$, $b = 0.05$, 95% CI = [0.02, 0.08]), suggesting that participants were more persuaded by the LLM-generated messages if they supported the policies pre-treatment more. The pre-treatment policy support × Human condition ($t(3209) = 0.02$, $p = 0.988$, $b = 0.00$, 95% CI = [−0.03, 0.03]) and the pre-treatment policy support × Human-in-the-Loop condition interaction effects were not significant ($t(3209) = 0.73$, $p = 0.468$, $b = 0.01$, 95% CI = [−0.02, 0.04]). Thus, in line with prior research[27], pre-existing identities and attitudes can play a crucial role in the effectiveness of persuasive messages. See "Moderators of the

Persuasion Effects" in Supplementary Methods (and esp. Supplementary Table 7) for a full description of all moderation analyses and results.

## Policy evaluation

We also asked participants to evaluate the messages on three important dimensions: how smart, compassionate, and ethical they perceived the policies to be. The effects of the messages on these evaluations of the policies were in line with the patterns of effects for overall policy support reported above. Aggregating data from Studies 1 and 2, we found that exposure to LLM-generated messages led participants to consistently evaluate the policies more favorably. Participants assigned to the LLM condition evaluated the policies as smarter ($t(3210) = 3.95$, $p < 0.001$, $b = 2.91$, 95% CI = [1.47, 4.36]), more compassionate ($t(3209) = 4.29$, $p < 0.001$, $b = 4.70$, 95% CI = [2.55, 6.84]), and more ethical ($t(3210) = 3.09$, $p = 0.002$, $b = 3.11$, 95% CI = [1.14, 5.08]) than did participants assigned to the Control condition. Additionally, exposure to LLM-generated messages led participants to evaluate the policies as positively as participants exposed to human-generated messages did. Specifically, participants assigned to the Human condition evaluated the policies as smarter ($t(3210) = 3.81$, $p < 0.001$, $b = 2.80$, 95% CI = [1.36, 4.24]), more compassionate ($t(3209) = 4.21$, $p < 0.001$, $b = 4.59$, 95% CI = [2.45, 6.72]), and more ethical ($t(3210) = 3.15$, $p = 0.002$, $b = 3.15$, 95% CI = [1.19, 5.11]) than participants in the Control condition. There were no significant differences in participants' evaluations of the policies between the LLM and Human conditions (smart: $t(2395) = 0.15$, $p = 0.881$, $b = 0.12$, 95% CI = [−1.40, 1.63], BF01 = 48.44; compassionate: $t(2395) = 0.10$, $p = 0.921$, $b = 0.11$, 95% CI = [−2.05, 2.27], BF01>>10,000; ethical: $t(2396) = −0.04$, $p = 0.970$, $b = −0.04$, 95% CI = [−2.03, 1.95], BF01>>10,000).

Finally, participants in the Human-in-the-Loop condition evaluated the messages more favorably compared to the Control condition (smart: $t(3210) = 5.20$, $p < 0.001$, $b = 3.89$, 95% CI = [2.42, 5.36]; compassionate: $t(3209) = 3.15$, $p = 0.002$, $b = 3.49$, 95% CI = [1.32, 5.67]; ethical: $t(3210) = 2.81$, $p = 0.005$, $b = 2.87$, 95% CI = [0.87, 4.87]). There were no significant differences in participants' evaluations of the policies between the Human-in-the-Loop and the LLM conditions (smart: $t(1570) = 1.32$, $p = 0.188$, $b = 1.01$, 95% CI = [−0.50, 2.52], BF01 = 16.66; compassionate: $t(1570) = −1.04$, $p = 0.299$, $b = −1.13$, 95% CI = [−3.27, 1.01], BF01 = 23.10; ethical: $t(1571) = −0.18$, $p = 0.859$, $b = −0.18$, 95% CI = [−2.15, 1.79], BF01 = 39.07), nor between the Human-in-the-Loop and the Human conditions (smart: $t(2395) = 1.43$, $p = 0.154$, $b = 1.12$, 95% CI = [−0.42, 2.65], BF01 = 15.56; compassionate: $t(2395) = −0.95$, $p = 0.343$, $b = −1.06$, 95% CI = [−3.25, 1.13], BF01 = 25.29; ethical: $t(2396) = −0.27$, $p = 0.791$, $b = −0.27$, 95% CI = [−2.29, 1.75], BF01 = 38.20).

See "Analysis of Other Post-Treatment Measures" in Supplementary Methods (and esp. Supplementary Table 4) for a full description of results predicting policy support evaluation by condition in Studies 1 and 2 combined.

## Perceived authorship

We also explored whether participants who read LLM-generated messages knew that they were reading LLM-generated messages in our research, as recent findings suggest LLM-generated messages are often perceived to be human messages[28]. In Study 2, we asked participants about who most likely wrote the messages with the options of "An adult person", "A group of people", "An expert on the topic", "An artificial intelligence program", "An intelligent adolescent", "An elementary school-age child", and "Other (please be specific)". Across conditions, the vast majority of participants believed that the messages were written by humans (i.e., options other than "An artificial intelligence program", LLM condition = 94.4%; Human condition = 94.7%; Human-in-the-Loop condition = 92.3%).

**Perceptions of Message Authors**

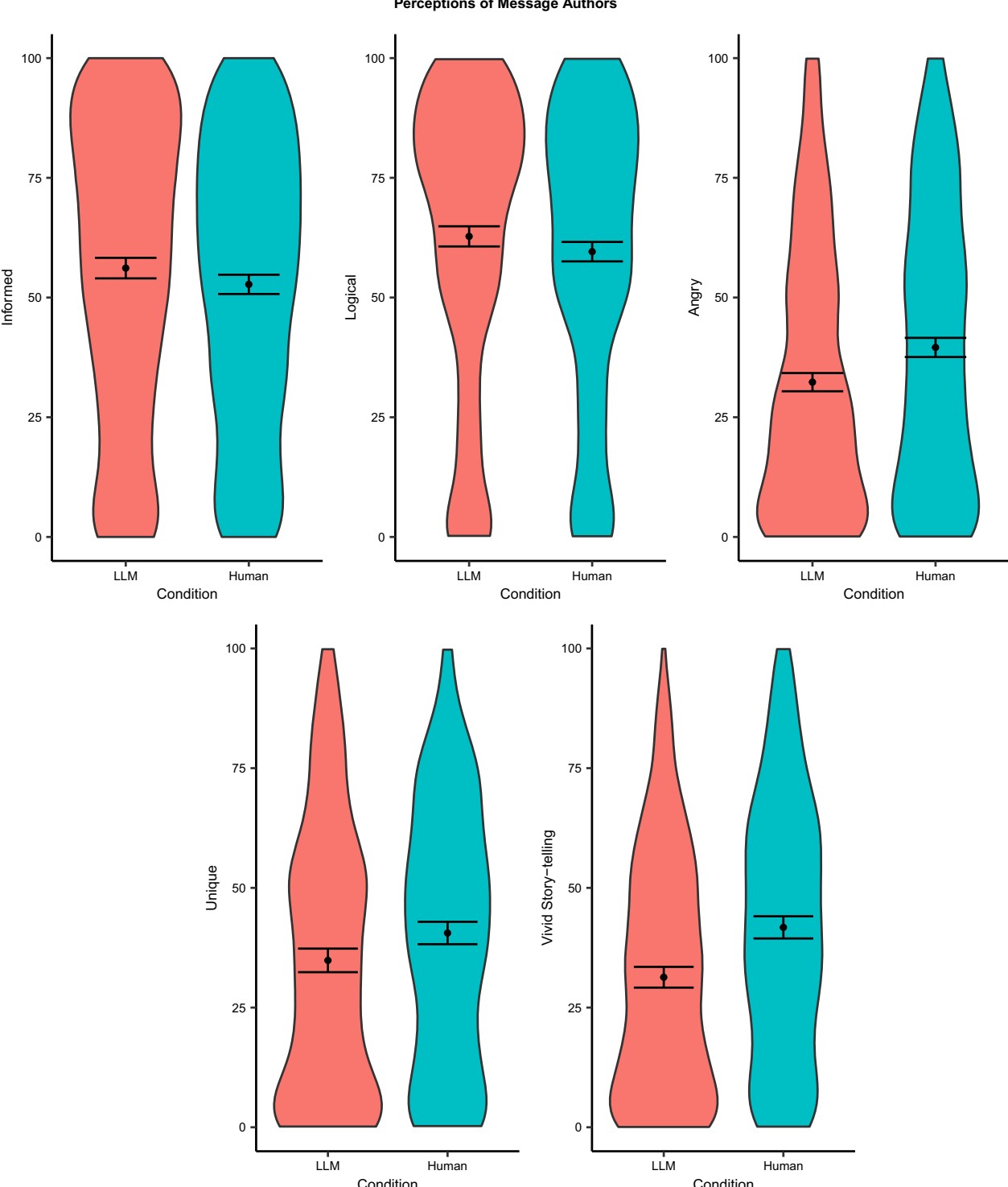

**Fig. 2 | Participants' perceptions of the author by condition in studies 1 and 2.** Data are analyzed using regression, two-tailed, and results are presented as mean values with 95% confidence intervals. Perceptions of informed, logical, and angry were asked in both studies, whereas unique and vivid story-telling was only assessed in Study 2. $N_{Studies\ 1\ and\ 2}$ = 3219 participants.

## Perceptions of authors of the messages

To understand the mechanisms driving the persuasive effects of human- and LLM-generated messages, we also investigated whether authors of human- and LLM-generated messages were perceived differently. Although participants who read messages generated by LLMs generally assumed these messages were human-generated, they nonetheless perceived authors of LLM-generated messages differently than authors of

human-generated messages. As Fig. 2. shows, participants in the LLM condition, compared to participants in the Human condition, rated the authors as better informed ($t$(1630) = 3.12, $p$ = 0.002, $b$ = 3.62, 95% CI = [1.34, 5.90]), more logical ($t$(1630) = 3.18, $p$ = 0.001, $b$ = 3.47, 95% CI = [1.33, 5.61]), less angry ($t$(1629) = −5.26, $p$ < 0.001, $b$ = −7.25, 95% CI = [−9.95, −4.55]), smarter ($t$(1629) = 2.04, $p$ = 0.042, $b$ = 2.21, 95% CI = [0.08, 4.34]), less unique ($t$(997) = −2.70, $p$ = 0.007, $b$ = −3.91, 95% CI = [−6.75,

−1.07]), and less inclined to use story-telling ($t(996) = −6.13$, $p < 0.001$, $b = −9.44$, 95% CI = [−12.46, −6.42]). We found no significant differences in participants' ratings of authors in the LLM and Human conditions on a number of other dimensions: assertive (BF01 = 31.61), moral (BF01 = 40.42), warm (BF01 = 23.69), authentic (BF01 = 6.29), creative (BF01 = 25.09), and interesting (BF01 = 19.49). See Supplementary Table 5 ($ps > 0.05$ for all). We also conducted a robustness check using a Bonferroni–Holm correction for these twelve analyses. Accordingly, all results are robust, except that different perceptions of how "smart" the authors of the AI-generated (versus human-generated) messages were seen to be became non-significant ($p = 0.504$) and "unique" became marginally significant ($p = 0.084$) and therefore these results are not visualized in Fig. 2.

We further conducted mediation analyses and found some evidence that each of the five perceptions of authors for which we found significant differences were associated with persuasion effects (see "Mediators of the Persuasion Effects" in Supplementary Methods). Importantly, these results should be interpreted with caution, as statistical mediation analysis is limited in its ability to identify causal mechanisms[29]. These analyses suggest that, while LLM- and human-generated messages led to similar levels of attitude change, they may have done so through at least somewhat distinct pathways. The attitude change caused by exposure to LLM-generated messages (relative to human-generated messages) may be driven in part by the perceptions that the author used more facts and evidence (indirect effect = 0.40, 95% CI = [0.07, 0.74]), logical reasoning (indirect effect = 0.38, 95% CI = [0.03, 0.72]), and a dispassionate voice (less angry; indirect effect = 0.28, 95% CI = [0.12, 0.47]), consistent with past findings that messages using more logic-related words tend to be more persuasive[30,31]. By contrast, the persuasiveness of human-generated messages (relative to LLM-generated messages) may be driven in part by perceptions of the message's author as unique (indirect effect = −0.60, 95% CI = [−1.01, −0.24]) and original (indirect effect = −0.72, 95% CI = [−1.21, −0.35]). See Supplementary Table 8a for full results of mediation analyses.

### Linguistic features of the messages

Additionally, we analyzed linguistic features of the LLM and human messages using dictionaries from the software Linguistic Inquiry and Word Count 2022[32]. These analyses provide insights into whether LLM and human-generated messages are linguistically distinguishable. Because we analyzed many linguistic features, we corrected $p$-values for multiple comparisons in all analyses below using the Benjamini-Hochberg correction. See Supplementary Table 6 for full results of analyses of linguistic features.

In terms of structural language variables, LLM-generated messages had a higher percentage of longer words (i.e., more than 7 letters) than human-generated messages ($t(198) = 6.31$, $p < 0.001$, LLM mean = 25.58% vs. human mean = 21.50%, $d = 0.89$, 95% CI = [2.80, 5.36]). LLM- and human-generated messages did not significantly differ in their number of words per sentence ($t(198) = −1.91$, $p = 0.990$, LLM mean = 18.71 vs. human mean = 19.9, $d = −0.27$, 95% CI = [−2.41, 0.04]), though LLMs produced longer messages overall ($t(198) = 4.53$, $p < 0.001$, LLM mean = 237.08 vs. human mean = 200.07, $d = 0.64$, 95% CI = [20.88, 53.14]). Using an open-source Python codebase[33], we estimated the Flesch-Kincaid reading level for the messages from both LLM- and human-generated messages, finding that LLM-generated messages featured higher grade-level vocabulary, on average ($t(151.26) = 2.22$, $p = 0.028$, LLM mean = 12.26 vs. human mean = 11.14, $d = 0.31$, 95% CI = [0.12, 2.12]). We also found language used in LLM-generated messages reflected more analytic thinking (as a metric of logical, formal reasoning; $t(198) = 9.48$, $p < 0.001$, LLM mean = 82.82 vs. human mean = 58.85, $d = 1.34$, 95% CI = [18.98, 28.95]), and less authenticity (perceived honesty, genuineness; $t(198) = −9.72$, $p < 0.001$, LLM mean = 7.48 vs. human mean = 29.96, $d = −1.38$, 95%

CI = [−27.03, −17.91]). LLM-generated messages also used fewer first-person singular pronouns (e.g., "I") than human-generated messages ($t(198) = −6.34$, $p < 0.001$, LLM mean = 0.01 vs. human mean = 0.66, $d = −0.90$, 95% CI = [−0.85, −0.45]), suggesting humans were more likely to speak from a personal perspective, and to invoke personal views and experiences. On the other hand, LLM-generated messages were more likely to use first-person plural pronouns (e.g., "we"; $t(198) = 4.29$, $p = 0.002$, LLM mean = 2.57 vs. 1.47, $d = 0.61$, 95% CI = [0.59, 1.60]), suggesting a higher rate of appeals to collective action and the common good. Interestingly, humans were more likely than LLMs to use third-person plural pronouns ("they"; $t(198) = −4.86$, $p < 0.001$, LLM mean = 1.00 vs. human mean = 1.83, $d = −0.69$, 95% CI = [−1.17, −0.49]) and negations ($t(198) = −9.0$, $p < 0.001$, LLM mean = 0.78 vs. human mean = 1.93, $d = −1.27$, 95% CI = [−1.41, −0.90]). In terms of overall emotional tone, however, we did not find a significant difference between LLM and human language ($t(198) = 1.06$, $p = 0.99$, LLM mean = 17.42 vs. human mean = 14.09, $d = 0.15$, 95% CI = [−2.85, 9.5]). Overall, these results suggest LLM-generated messages are on average more technical and analytical than human-generated messages and employ a more sophisticated vocabulary. In contrast, human-generated messages were more likely to employ personal appeals.

## Discussion

Across three pre-registered experiments, we provide evidence that exposure to LLM-generated messages can shift humans' support for policies to align more with the position advocated for in the messages. We find such effects across different policies, including several highly polarized policies (e.g., an assault weapon ban) and a less polarized policy (a public smoking ban), using national samples that are representative of the US population across three benchmarks (gender, race/ethnicity, and age). Although the effects of LLM-generated messages are small, they are comparable to the magnitude of effects found in prior persuasion research[8,9].

### LLM versus human messages

In both studies featuring LLM- and human-generated messages, we found the influential effects of these two kinds of messages were comparable in size, suggesting LLMs may have already caught up to the persuasive capacity of everyday people, a critical benchmark of human-like performance. There may still be conditions where LLM-generated messages are more persuasive than human-generated messages and vice versa. The human-generated messages tested in our studies were written by laypeople, as opposed to political consultants or professional campaigners. Expert humans may produce more persuasive appeals than LLMs, at least at present. As a result, LLM-generated messages might be used disproportionately by campaigns with fewer resources. Future research is needed to compare the persuasiveness of more sophisticated LLMs to the persuasiveness of human experts. Since the conclusion of our research, more powerful LLM models have been developed and become accessible to the public. We compared Study 1's messages generated using GPT 3 on December 3, 2022, with messages generated using GPT 4 ("ChatGPT") on November 8, 2023, using the same prompt. We found there is a great deal of linguistic similarity between these LLMs (see "Comparison of GPT 3 and GPT 4 Messages" in Supplementary Methods). As such, the effects we report in the main text will likely persist if more recent models, like GPT 4 or 4o, are used for generating messages.

In an internal meta-analysis of Studies 1 and 2, we tested for a possible difference in the persuasiveness of the messages in the Human-in-the-Loop condition relative to the messages in the LLM condition ($t(1571) = 1.94$, $p = 0.052$, $b = 0.94$, 95% CI = [−0.01, 1.88]). While recent research suggests laypeople have limited abilities to accurately forecast what are more versus less persuasive political appeals[34,35], future research should investigate this topic further. More

generally, many variables may influence whether and when LLM-generated messages may outperform human-generated messages (or vice versa), including the data used to train LLMs, human message authors' expertise, and message recipients' prior views about the topic of persuasion. Future research is needed to investigate these and other potential moderating factors.

Despite the similar effect sizes of LLM-generated and human-generated messages, we found suggestive evidence that the mechanisms underlying the effects might be different. The authors of LLM-generated messages were perceived to be more logical, better informed, and less angry than the authors of human-generated messages. In contrast, the authors of human-generated messages were perceived to be more unique and to use more vivid story-telling than the authors of LLM-generated messages. Mediation analyses suggest that each of these five perceptions may be involved in the persuasion effects. This nuanced understanding of AI and human communication styles not only enriches our results but also offers valuable insights into effective message crafting in different contexts. For instance, utilizing LLM-generated arguments might prove more persuasive for audiences that value rational and evidence-based appeals, while human-generated arguments could be more impactful when addressing a relatable audience in a personal and intimate setting. Future research is needed to test these possibilities empirically.

LLM-generated and human-generated messages were not only perceived to be different. Text analysis revealed several linguistic differences. LLM-generated messages were on average more technical and analytical than human-generated messages and used a more sophisticated vocabulary. In contrast, human-generated messages used more pronouns and relied more on personal appeals[31].

While participants in our study were not able to leverage these differences to identify that the LLM-generated texts were authored by AI, these text and perception differences suggest humans might be able to learn to identify LLM-generated text. It is notable, however, that on many other dimensions, LLM-generated messages did not diverge from human-generated messages. For example, authors of the two types of messages were similarly likely to be seen as warm ($BF01 = 23.69$), compassionate ($BF01 >> 10,000$), and creative ($BF01 = 25.09$). Further, recent research suggests LLMs are readily capable of increasing their use of language markers of human authors, such as pronouns[28].

Finally, a potential alternative mechanism for how LLM-generated and human-generated messages led to attitude change is that they did so through demand effects. If demand effects are the main mechanism, these effects are more likely to generalize to natural contexts that also clearly signal the message source's private views (e.g., internet campaign ads and direct mail). However, recent research finds demand effects do not usually occur in online survey experiments such as ours[36], suggesting it is unlikely that our findings were driven by demand effects alone, if any. Nonetheless, future research should further assess the role of demand effects in persuasion experiments such as ours.

## Impact statement
Future applications of LLM-generated persuasive political messages are likely to be numerous and varied, and, correspondingly, their societal impacts of specific applications could be benevolent, pernicious, or inconsequential. Among benevolent applications, LLMs could be used to help people construct effective political appeals for their political views, empowering citizens who may currently lack such capacity, and potentially democratizing political discourse. Among more neutral applications, the persuasive capacity of LLM-generated content could be used by people working in politics to more efficiently generate persuasive text that they would generate anyway—e.g., speeches, website content, slogans, canvassing scripts—though without fundamentally changing these activities and the content of

political communications. Among pernicious use cases, however, LLMs could be abused to impersonate a large number of individuals in political discourse, inundating the landscape of political communication both online (e.g., via social media, in discussion fora, and via comments on news articles) and off-line (e.g., via peer-to-peer text messages, email, direct messages, letters to newspapers[37] and letters to elected officials). At scale, such efforts could easily distort democratic processes. In fact, scholars have already raised concerns that LLMs could be used by domestic and foreign actors to deceive the electorate, producing convincing and tailored misinformation for specific populations or platforms at a massive scale[38]. Additionally, such efforts could distort politicians' understanding of the attitudes and preferences of the mass public, for example, if LLMs were used to create messages to elected representatives ostensibly written by constituents.

To be clear, for ethical reasons, we did not directly test whether LLM-generated misinformation could persuade people. However, given our demonstration that LLMs are capable of political persuasion, and the open availability of these models, including ones with few guardrails, the application of LLMs to produce persuasive content featuring misinformation and/or misleading information is very likely now possible. Consistent with this concern, research shows LLM-generated propaganda with misleading and/or inaccurate content can influence individuals' views[39]. Future research is critical to understand these evolving capacities, as well as what can be done to address them[40].

The effects of these applications could be further amplified by LLMs' abilities to customize messages based on the characteristics of recipients targeted for persuasion. Recent work reveals that LLMs can be used to generate customized messages to micro-target different recipients, though evidence on the extent of their persuasive effects remains mixed. For example, one study[14] found that messages custom-generated by LLMs for recipients with different traits and political ideology are perceived to be more persuasive by perceivers with the target traits and beliefs than by perceivers without them. However, another study[41] found that messages custom-generated by LLMs were no more effective in increasing support for policies than a generic message that was generated with a sophisticated prompt, but not customized for a specific target audience. These inconsistencies in the effects of micro-targeting using LLMs suggest continued research is needed to track the potential impact of this application.

Taken together, our findings demonstrating the potential use of LLMs for political persuasion should engender urgency among policymakers considering what uses of LLMs to regulate, and stimulate further discussion and research on effective regulation. At this stage, it is difficult to confidently recommend public policy to respond to this new capacity of AI, in part because it is unclear how it will be used, and which uses will have negative effects on society. However, analysts should closely track applications of AI to political persuasion, ideally employing technology that can reliably identify LLM-generated text content.

Potential policy responses could include mandatory disclosure of LLM-generated political content, embedding digital watermarks in LLM-generated political appeals, training AI models to detect LLM-generated political appeals for potential regulation on social media platforms or discussion fora, and designing extensive guardrails on publicly available LLMs to refuse questionable requests to generate persuasive political content, particularly requests generating messages supporting false political claims (c.f.[42]). Additionally, digital literacy education programs should integrate AI literacy into their curricula to foster critical and discerning engagement with persuasive content[43]. Media companies may adopt more robust fact-checking processes for highly persuasive or influential content, particularly those bearing hallmarks of AI generation. It is important for future research to explore the feasibility and efficacy of these and other

potential interventions to counteract potential negative effects of LLM-generated political messages.

## Methods

All studies were approved by the Institutional Review Board at Stanford University (Protocol ID: 32506). All participants provided informed consent and were paid a small monetary fee for their participation. Participants were randomly assigned to their condition and blind to the study design. We did not use deception. Because the studies were conducted online, there was no interaction between the experimenter and participants. We followed pre-registered protocols that are publicly available at osf.io/7wyeh for Study 1 (November 14, 2022), osf.io/jx6dp for Study 2 (December 5, 2022), and osf.io/eczh3 for Study 3 (December 24, 2022). There was no deviation from our analysis plan for any of our three studies. We used the survey software platform Qualtrics to design and collect data for all three studies, files for which are publicly available at osf.io/8yxvr. For a full description of measures summarized below, see "Measures for Studies 1, 2, and 3" in Supplementary Methods.

For all studies, gender was determined based on self-identification of participants. We did not consider or pre-register any analyses disaggregating the data by participant's sex or gender, as it was not a theoretically relevant variable to our research questions and recent research finds the persuasive effects of political messages are similar for participants of different genders[44].

### Study 1 participants

We recruited a convenience sample of US participants from Prolific.com. We pre-registered seeking a sample size of 1200 participants (300 participants per condition). Our final sample was 1203. Sensitivity power analyses suggest that this sample size provided 95% power (alpha = 0.05) to detect a treatment effect of $b = 2.85$.

A total of 2096 participants responded to our recruitment advertisement. Before treatment assignment, we excluded participants (i) who had missing values for any item making up the pre-treatment dependent variable (i.e., support for a smoking ban), (ii) who already reported a very high level of support for the smoking ban (response to the composite score for the smoking ban of more than 95), (iii) who failed the attention check, (iv) who dropped out of the study before treatment assignment, and (v) who indicated that they were under 18 years old. After treatment assignment, we excluded from our analysis those (vi) who had missing values for any item making up the post-treatment dependent variable (i.e., support for a smoking ban) and (vii) with the same Participant ID by keeping only the first case, leaving us with a final sample of 1,203. Among them, 567 self-identified as female and 631 as male. The mean age was 38.37 ($SD = 13.12$). 632 participants reported having a college-degree. 927 identified as White/Caucasian, 75 as Black/African American, 85 as Asian/Asian American, 69 as Latino/Hispanic, 42 as "other". 551 participants identified as Democrats, 219 as Republicans, and 377 as Independents or other. 51 participants reported having no political affiliation. Demographic information from 5 participants is missing.

### Study 1 procedure

Study 1 employed a 2 (Time: Pre-treatment vs Post-treatment) × 4 (Message: LLM vs Human vs Human-in-the-Loop vs Control) within-between-subject design[45]. First, participants completed a short questionnaire, including the pre-treatment measure of support for a smoking ban, demographic questions, and an attention check.

Next, participants were randomly assigned to one of four message conditions. In the LLM condition, participants read an AI-generated message. In the Human condition, participants read a human-generated message. In the Human-in-the-Loop condition, participants read an AI-generated message that was selected by a human. In

the Control condition, participants read a human-generated message on an unrelated topic. For more details, please see the "Messages" section below.

Finally, participants completed a post-treatment measure of support for a smoking ban. Participants also completed a series of other measures for exploratory analyses.

### Study 1 messages

For all experimental conditions, messages were generated with the aim of persuading readers to support a smoking ban in public places. For the LLM condition, 50 messages were generated by GPT-3, an AI program (Text-Davinci-002 model) on October 26, 2022. We used the following prompt: "Please try your best to write a message of about 200 words that can persuade a reader to agree with the following idea. 'We should enforce a total smoking ban in public places.'" The average number of words for the 50 messages was 188.9. Participants in this condition were randomly assigned to read one of the 50 messages. For the Human condition, 50 messages were generated by human participants (recruited from Prolific.com). We used the same prompt as for the LLM condition. In addition, we incentivized human writers with the following instructions: "Note that we will actually present the message you write to a future participant and see if they report an increase in their level of support for the smoking ban after reading the message. We will give a $100 bonus to the person whose message is most persuasive. To give yourself the greatest chance of winning the bonus, please write the message you believe will be most persuasive to a future participant." Participants had to write at least 125 words to submit their responses. The average number of words for the 50 messages was 158.2. Participants in this condition were randomly assigned to read one of the 50 messages. For the Human-in-the-Loop condition, 300 human participants each reviewed five AI-generated messages (randomly selected from the pool of 50 GPT-3-generated messages) and selected the one that they thought was most likely to succeed in persuading a recipient to send to a future participant. Only individuals whose own pre-treatment support for the smoking ban was 60 or greater were allowed to be a message writer or a curator (see "Measures" below). In the Control condition, participants were randomly assigned to one of three human-generated messages on an unrelated topic (residential mobility, the history of skiing, or event licensing in a midsize town). Messages for all conditions can be found at osf.io/8yxvr.

### Study 1 main outcome

Both pre- and post-treatment policy support was measured across five items (e.g., "Do you support or oppose a total smoking ban in all public spaces?") on 101-point scales from 0 to 100 and reverse-coded as necessary. A composite score for policy support was calculated as the mean of all five items with higher scores indicating stronger support for the policy (Cronbach's $\alpha = 0.95$ pre-treatment, Cronbach's $\alpha = 0.96$ post-treatment). We used a 101-point scale because we anticipated small treatment effects. Such granular measures allow for capturing more nuanced differences[46] and previous research suggests that 101-point scales allow for detecting small treatment effects in survey experiments[47].

### Study 1 other outcomes

Following treatment, message recipients were asked to rate their level of agreement with a series of statements pertaining to how smart, logical, empathetic, compassionate, moral, and ethical the idea of a total smoking ban is (e.g., "How much do you agree with the following statements?: A total smoking ban is a smart idea"). Responses were recorded on 101-point scales from 0 to 100, with higher scores indicating greater agreement (see "Analyses of Other Post-Treatment Measures" in Supplementary Methods for detailed descriptions of how these items were formed into composites and subsequently analyzed).

Message recipients were also asked to rate how well the following words described the author of the message, each on a 101-point scale from 0 to 100, with higher scores indicating greater perceived descriptive accuracy: smart, intelligent, compassionate, empathetic, warm, cold, pushy, angry, logical, moral, ethical, factual, well-informed (see "Author Perception Analyses" in Supplementary Methods for detailed descriptions of how these items were formed into composites and subsequently analyzed).

## Study 1 pre-registered analytic strategy

As pre-registered, we estimated three regression models (see Supplementary Table 1a for a detailed summary of results). In the first regression, the post-treatment measure of policy support was regressed on the dummy-coded variables for the three experimental conditions (LLM, Human, and Human-in-the-Loop, with the Control condition as the reference category) while controlling for the pre-treatment measure of the smoking ban support variable. In the second regression, we re-ran the first regression model, dropping the Control condition and using the Human condition as the reference category. In the third regression, we reran the first regression model, dropping the Control condition and the Human condition and using the LLM condition as the reference category.

## Study 1 robustness check

For robustness, a parallel set of regression models were estimated in which the dependent variable is the change in support (the difference between the post-treatment policy support and the pre-treatment policy support) and the pre-treatment policy support score is not used as a control variable. The results were very similar to our pre-registered regression models (see Supplementary Table 1b for a summary).

## Study 2 participants

We recruited a politically-balanced sample of US participants from Prolific.com and CloudResearch. We pre-registered seeking a sample size of 2000 participants (500 participants per condition). Our final sample was 2016. Sensitivity power analyses suggest that this sample size provided 95% power (alpha = 0.05) to detect a treatment effect of $b = 1.87$.

A total of 3541 participants responded to our recruitment advertisement. Before treatment assignment, we excluded participants (i) who had missing values for any item making up the pre-treatment dependent variable (i.e., support for an assault weapon ban), (ii) who already reported a very high level of support for an assault weapon ban (response to the composite score for an assault weapon ban of more than 95), (iii) who failed the attention check, (iv) who dropped out of the study before treatment assignment, and (v) who indicated that they were under 18 years old. After treatment assignment, we further excluded from our analysis those (vi) who had missing values for any item making up the post-treatment dependent variable (i.e., support for an assault weapon ban), and (vii) with the same Participant ID by keeping only the first case, leaving us with a final sample of 2016 participants. Among them, 987 self-identified as female, 995 as male, 11 as non-binary, and 23 declined to report their gender. The mean age was 40.31 (SD = 13.73). 1124 participants reported having a college-degree. 1,530 identified as White/Caucasian, 161 as Black/African American, 69 as Latino/Hispanic, 135 as Asian/Asian American, 83 as being mixed-race, and 36 as "other" (2 participants declined to report their race). 780 participants identified as Democrats, 760 as Republicans, and 476 as Independents.

## Study 2 procedure

Like Study 1, Study 2 employed a 2 (Time: Pre-treatment vs Post-treatment) × 4 (Message: LLM vs Human vs Human-in-the-Loop vs Control) within-between-subject design[45]. First, participants completed a short questionnaire, including the pre-treatment measure of

support for an assault weapon ban, demographic questions, and an attention check.

Next, participants were randomly assigned to one of four message conditions. In the LLM condition, participants read an AI-generated message. In the Human condition, participants read a human-generated message. In the Human-in-the-Loop condition, participants read an AI-generated message that was curated by humans. In the Control condition, participants read a human-generated message on an unrelated topic. For more details, please see the "Messages" section below.

Finally, participants completed a post-treatment measure of support for an assault weapon ban. Participants also completed a series of other measures for exploratory analyses.

## Study 2 messages

Similar to Study 1, we generated all messages that participants read in the three experimental conditions and the Control condition. For all experimental conditions, messages were generated with the aim of persuading readers to support an assault weapon ban. For the LLM condition, 50 messages were generated by GPT-3.5 (Text-Davinci-003 model) on December 3, 2022. We used the following prompt: "Please try your best to write a message of about 250 words that can persuade a reader to agree with the following idea. 'We should enforce an assault weapon ban.'" The average number of words for the 50 messages was 285.22. Participants in this condition were randomly assigned to read one of the 50 messages. For the Human condition, 50 messages were generated by human participants (recruited from Prolific.com). We used the same prompt as for the LLM condition. Human writers were incentivized with the same instructions as in Study 1. Participants had to write at least 200 words to submit their responses. The average number of words for the 50 messages was 241.92. Participants in this condition were randomly assigned to read one of the 50 messages. For the Human-in-the-Loop condition, 300 participants each reviewed five AI-generated messages (randomly selected from the pool of 50 GPT-3-generated messages) and selected the one that they thought was most likely to succeed in persuading a recipient to send to a future participant. Like in Study 1, only individuals whose own support for the assault weapon ban was 60 or greater were allowed to be a message writer or curator. In the Control condition, participants were randomly assigned to read one of the same three human-generated messages on an unrelated topic. Messages for all conditions can be found at osf.io/8yxvr.

## Study 2 main outcome

Both pre- and post-treatment policy support was measured in a similar way as in Study 1 but focused on a different policy (an assault weapon ban instead of a smoking ban). A composite score for policy support was calculated in a similar manner to Study 1 (Chronbach's $\alpha = 0.95$ pre-treatment, Chronbach's $\alpha = 0.98$ post-treatment).

## Study 2 other outcomes

Following treatment, similar to Study 1, message recipients were asked to rate their level of agreement with a series of statements about how smart, logical, empathetic, compassionate, moral, and ethical the idea of an assault weapon ban is. Responses were recorded on 101-point scales from 0 to 100, with higher scores indicating greater agreement. See "Analyses of Other Post-Treatment Measures" in Supplementary Methods for detailed descriptions of how these items were formed into composites and subsequently analyzed.

Additionally, message recipients were asked to rate how well a series of statements (e.g., "Used a lot of facts and evidence", "Was very well-informed") and adjectives (e.g., "Empathetic", "Moral", "Authentic") described the message author on a 101-point scale ranging from 0 ("Not at all") to 100 ("A great deal"). See "Author Perception Analyses" in Supplementary Methods for detailed

descriptions of how these items were formed into composites and subsequently analyzed.

Message recipients were also asked in an open-ended, text-response question how they would describe the author of the message they read and if they had any questions about them.

Finally, message recipients were asked to assess who was most likely to have authored the message they read ("An adult person", "A group of people", "An expert on the topic", "An artificial intelligence program", "An intelligent adolescent", "An elementary school-aged child", or "Other (please specify)").

### Study 2 pre-registered analytic strategy

As pre-registered, we estimated three regression models identical to those of Study 1 (see Supplementary Table 2a for a detailed summary of results).

### Study 2 robustness check

For robustness, a parallel set of regression models was estimated in a manner similar to the robustness check for Study 1(results are very similar to our pre-registered regression models, as summarized in Supplementary Table 2b).

### Study 3 participants

We recruited US. participants representative of US gender, race, and age demographics from Prolific.com. We pre-registered seeking a sample size of 1600 participants (800 participants per condition). Our final sample was 1610. Sensitivity power analyses suggest that this sample size provided 95% power (alpha = 0.05) to detect a treatment effect of $b = 1.23$.

A total of 1795 participants from Prolific.com responded to our recruitment advertisement. Before treatment assignment, we excluded participants (i) who had missing values for any item making up the pre-treatment dependent variable (i.e., policy topic; see "Procedure" below), (ii) who failed the attention check, (iii) who dropped out of the study before treatment assignment, (iv) and who indicated that they were younger than 18 years old. After treatment assignment, we further excluded from our analysis those (v) with the same Participant ID by keeping only the first case. Because none of these remaining participants had missing values for any item making up the post-treatment dependent variable, we did not further exclude any participants from our analysis, leaving us with a final sample of 1610. 828 self-identified as female and 782 as male. The mean age was 44.04 ($SD = 15.37$). 876 participants reported having a college degree. 1233 identified as White/Caucasian, 187 as Black/African American, 85 as Latino/Hispanic, 80 as Asian/Asian American, and 25 as "Other". There were 809 who self-reported as Democrats, and 333 who self-reported as Republicans. The remaining participants self-reported as Independents and others.

### Study 3 procedure

Study 3 followed a similar procedure to Studies 1 and 2 except that Study 3 only incorporated an LLM and a Control condition. Specifically, Study 3 employed a 2 (Time: Pre-treatment vs Post-treatment) × 2 (Message: LLM vs Control) × 4 (Topic: Carbon Tax vs Child Tax Credit vs Parental Leave Program vs. Automatic Voter Registration) within-between-between-subject design. First, participants completed a short questionnaire, including demographic questions, an attention check, and the pre-treatment measure of support for the policy topic they were randomly assigned to.

Next, participants were randomly assigned to one of two message conditions. In the LLM condition, participants read an AI-generated message. In the Control condition, participants read a human-generated message on an unrelated topic. For more details, please see the "Messages" section below.

Finally, in a similar manner to Studies 1 and 2, participants completed a post-treatment measure of policy support.

### Study 3 messages

We generated all messages that participants read in the experimental condition and the Control condition. For the experimental condition, messages were generated with the aim of persuading readers to support one of four issue topics: a carbon tax, paid parental leave, a child tax credit, and automatic voter registration. For the LLM condition, 15 messages for each of the four topics were generated by GPT-3.5 (Text-Davinci-003 model) on December 8, 2022. The prompt for each issue topic began with the following instructions: "Please try your best to write a message of about 250 words that can persuade a reader to agree with the following idea." The remainder of each prompt varied by topic: for a carbon tax, "The US should have a federal carbon tax."; for paid parental leave, "The US federal government should fund paid parental leave."; for a child tax credit, "The US federal government should implement a child tax credit."; for automatic voter registration, 'Eligible Americans should be automatically registered to vote.' The average number of words for the 15 messages for each topic was 282.95 for the carbon tax messages, 286.67 for the paid parental leave messages, 281.2 for the child tax credit messages, and 281.07 for the automatic voter registration messages. Participants in this condition were randomly assigned to read one of the 15 messages for the topic they were randomly assigned to. In the Control condition, participants were randomly assigned to one of four human-generated messages on an unrelated topic (three of which were conserved from Studies 1 and 2). Messages for all conditions can be found at osf.io/8yxvr.

### Study 3 main outcome

Both pre- and post-treatment policy support was measured similarly as in Studies 1 and 2, but focused instead on four different and distinct policies: a carbon tax, a child tax credit, a parental leave program, and automatic voter registration. A composite score for policy support was calculated in a similar manner to Studies 1 and 2 for each policy topic (for the carbon tax, Chronbach's $\alpha = 0.98$ pre-treatment, Chronbach's $\alpha = 0.98$ post-treatment; for the child tax credit, Chronbach's $\alpha = 0.97$ pre-treatment, Chronbach's $\alpha = 0.98$ post-treatment; for the parental leave program, Chronbach's $\alpha = 0.97$ pre-treatment, Chronbach's $\alpha = 0.98$ post-treatment; for the automatic voter registration, Chronbach's $\alpha = 0.98$ pre-treatment, Chronbach's $\alpha = 0.98$ post-treatment).

### Study 3 pre-registered analytic strategy

As pre-registered, we estimated a series of regression models in which post-treatment policy support was regressed on the treatment condition, controlling for pre-treatment policy support and policy topic (dummy-coded) (see Supplementary Table 3a for a detailed summary of results).

### Study 3 robustness check

For robustness, a parallel set of regression models was estimated in a manner similar to the robustness checks for Studies 1 and 2 (results are very similar to our pre-registered regression models, as summarized in Supplementary Table 3b).

### Reporting summary

Further information on research design is available in the Nature Portfolio Reporting Summary linked to this article.

## Data availability

Anonymized data generated in all studies have been deposited in the following OSF repository: https://doi.org/10.17605/osf.io/8yxvr.48 The raw study data are protected and are not available due to data privacy laws.

## Code availability

Analysis codes for all studies are available at https://doi.org/10.17605/osf.io/8yxvr[48]. All analysis codes written with RStudio 2022.07.2 Build

576. All packages used in statistical analysis are specified in individual analysis codes.

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

## Acknowledgements

The authors received funding from the Stanford Center on Philanthropy and Civil Society.

## Author contributions

H.B., J.G.V., J.C.E., and R.W. designed the research. H.B. conducted the research and analyzed the data. H.B., J.G.V., S.M., J.C.E., and R.W. wrote the paper. H.B. and R.W. are both corresponding authors.

## Competing interests

The authors declare no competing interests.
