## [Transparent Peer Review file · Nature Communications]

LLM-generated messages can persuade humans on policy issues

Corresponding Author: Dr Hui Bai

Version 0:

Reviewer comments:

Reviewer #1

(Remarks to the Author)

This manuscript provides a clear and straightforward experiment that shows that AI can write text that could be as persuasive as human-written text. Overall, it is a well-written manuscript with pretty robust results. I do have some minor concerns about the manuscript, detailed below, but I have some greater concerns about the novelty of the work, and would like the authors to produce a stronger novelty argument for publication in a top journal.

Generally, as noted above, I was very excited about this paper: although the setup is very simple, the novelty pitch was potentially interesting. However, to me, the findings -- including the core findings ("first evidence that LLM generated messages influence humans' support for policy issues") -- do not offer enough over the earlier work, most relatedly, the paper "Working With AI to Persuade: Examining a Large Language Model's Ability to Generate Pro-Vaccination Messages" (<https://dl.acm.org/doi/abs/10.1145/3579592>; I will refer to it here as the "vaccines paper"; it is not cited in the manuscript; it was written by members of the same Stanford institute). Can messages created by AI persuade? Yes, we already know that from the vaccines paper. Can they persuade about *policy issues*? Well if you consider vaccination attitudes "policy issues" (as I think one should), I believe the novelty of this current paper over the vaccine paper is again somewhat diminished. The vaccine paper even uses similar methods, first to evaluate the messages created (granted, not along the same dimensions, and they did not evaluate the policy attitudes) and then to compare their persuasion to human-created ones (again, granted, in a slightly different setup).

Taken together, I would like to see a much stronger novelty argument for this manuscript over this and other recent "AI-text persuasion" work. I would need a strong argument to be convinced this paper is appropriate for this venue; but that would be up to the editors to decide.

A more compelling path for a more significant contribution in this work is to develop the investigation of the persuasion aspect. Recent papers, for example, looked at the capabilities of LLM-based *personalization* to persuade (see "Evaluating the persuasive influence of political microtargeting with large language models", <https://osf.io/wnt8b/>, and "The Potential of Generative AI for Personalized Persuasion at Scale", <https://psyarxiv.com/rn97c>). These papers are new, and not yet (to my knowledge) peer reviewed. I do not expect the manuscript to refer to them, although that would be nice, for completeness, unless there is a reason not to. I am citing these papers here as examples of the type of novelty required in this area to justify a high-profile publication -- again, at least in my opinion.

These papers went further than investigating the text-creation capabilities of LLMs which, at this point, are well recognized. As the manuscript also notes, people cannot distinguish human- and AI-written text: so why would AI-written text be less persuasive? The interesting questions, to me, lie in the additional capabilities of LLMs: to produce and test messages cheaply and at scale.

A more minor contribution of the paper is showing that the AI-created text was different along several characteristics from human-created text. But it wasn't shown HOW that difference may have contributed to persuasion -- which would be a more significant contribution -- and the differences between AI and human text had been explored in other papers, e.g. "Human heuristics for AI-generated language are flawed" (cited in the manuscript as footnote 22).

Finally, the manuscript highlights in the discussion that "optimal persuasive messaging... involve leveraging the strengths of

artificial and human intelligence together." This is interesting but a little bit expected (I do not know if I would call it "intriguing"). A more direct comparison would look at human-selected AI messages compared to human-selected human messages. Actually, why not let AI pick the most promising messages? That would be interesting as a comparison too.

To me these are the main issues of this manuscript, touching on its novelty. There are other minor questions and places that require clarification that I list below, which I hope can be addressed with more careful reporting, listed next.

- I am afraid the very title of the work could be a bit misleading. The manuscript does not show that "Artificial Intelligence Can Persuade Humans on Political Issues"; it shows that "Text created by AI" can do the same. The difference is that people were not aware that they are getting AI-written arguments as suggested in the title (and in some of the manuscript text). As evident from the analysis, AI was not even a consideration for the participants (97% thought the writer was human, not surprisingly). That participants thought the text was from humans is totally fine – but that makes the title not aligned with the main investigation. There is some language like that throughout, which I encourage revisiting (in some cases it is more palatable than others).

- I wonder what can be said about the potential impact of demand effects here. "participants reported their support for a policy before and after reading the assigned message". The participants thus know what they are asked to provide, and understand that they are being nudged to change their view. Is that a concern? Do other persuasion papers control for that in any way -- and, was anything done in this research to control for it?

- Similarly, using a 101-point policy support scale(!) seems to exaggerate the effect size for the figures. Is using this scale a common practice in this kind of study? The manuscript (or SI) should justify the choice of the scale -- I do not think I saw any justification. I also do not believe the actual number of points on the scale was ever mentioned (in SI or in OSF, but I may have missed it). Could the score be anything from 0 to 100? Or did it have 4, 5, 7, 9, 11 points? (see <https://pubmed.ncbi.nlm.nih.gov/10769936/>). Relatedly: it would also be useful to see histograms for the key DVs in the SI.

- Speaking of data and distributions: the manuscript reported that "the issues are polarized" but how do we know that? What was the distribution of preliminary opinions in the data?

- I am confused about the reported numbers in the text, what tests/models they correspond to, and where do the numbers (e.g. "b = 3.62") come from. A short clarification may be needed.

- Also confusing: "LLM-generated messages were evaluated as positively as human-generated messages" - I think this refers to the evaluation of the policies, not the messages.

- The analysis of the author characteristics is under "Features of the Messages". Why? Also, was there a correction for performing multiple tests here?

- How participants thought AI was involved in the writing is an interesting question. The details of how this question was asked are a little hidden in the SI but I think the text should be clear that when "we asked participants about who wrote the messages" only one of the several options was AI (as a reader, I thought you asked the "AI or human" question as used in previous studies). Regardless, about the statement "LLMs can readily generate messages perceived to be human-generated at the same rate as actually human-generated messages" -- I know this is not the manuscript's main claim but it is well documented by now in many papers. The point made in the discussion ("guessing that authors of both human-generated and LLM-generated messages were human at almost identical rates (94.7% and 94.4%, respectively) is therefore less interesting: the participants had no reason to believe AI was involved.

- "LLM-generated messages used fewer pronouns, more long words" -- Why is that interesting? The manuscript does not set it up, or return to it in the discussion. (these results also align with previous work).

Finally, the manuscript could use some streamlining and tightening. Two examples:

- In the abstract, the sentence "Our results show LLMs can persuade humans, even on highly polarized policy issues" -- seems repetitive with the previous text.

- The intro can also be streamlined. When the manuscript gets to "whether LLMs are capable of persuading people to shift their views on political issues" it's been already mentioned several times.

Reviewer #2

(Remarks to the Author)

Summary

This paper reports a series of three survey experiments with US adults in which the persuasiveness of political messages generated by the large language models GPT3 and GPT3.5 is tested against the persuasiveness of messages generated by humans (human condition), messages generated by AI and selected by humans (human-in-the-loop), or a control condition (placebo message). The different experiments study different political issues. Across the three experiments and corresponding political issues, the aggregate result is clear: the AI generated messages (1) persuade people on average and (2) are approximately as persuasive as the human-generated messages and the human-in-the-loop messages. In other

words, the headline result is: yes, politically persuasive messages generated by GPT3 and 3.5 can exert a causal effect on US adults' political attitudes.

Comments

The contribution of this paper is clear and straightforward, and the headline result is convincing. Below I offer some observations on and suggestions that may improve the paper.

- I appreciated the balanced discussion of the possible positive/neutral/negative implications of AI-generated political communication throughout the paper. I have reviewed several related papers in recent months and too often it is assumed that the contribution of AI to political communication is necessarily going to be negative. This would not seem to be the case, and the current authors do a good job of offering a more balanced (and I think, accurate) perspective. They could perhaps go even further and explicitly say that persuasion is part and parcel of a healthy democratic politics — that is, people should not be concerned about persuasion per se. Rather, the concerns surround misinformation and misrepresentation embedded in the attempts at persuasion, as well as problems related to a small number of people “flooding the zone” with AI-generated content, as the authors note.
- I also appreciated the use of multiple issues across the experiments, which helps the generalizability of their estimates.
- I think the authors should explicitly discuss an important limitation of their human-generated messages: they were written by laypeople, not political consultants or professional campaigners. This has direct implications for how AI may be taken up by political campaigns. For example, while AI may match laypeople for persuasiveness, it may still be outperformed by professionals. In this case, larger campaigns that have substantial resources may still favor hiring human professionals. And AI may disproportionately be used by smaller campaigns, who lack the resources to hire the comparatively more expensive professionals.
- It would be nice to see covariate balance checks in the appendix for each study. The authors do a good job reporting post-treatment missingness in their outcome variables — and with the exception of study 1 it looks generally very low. Nevertheless, to reassure readers that randomization worked as intended, covariate balance checks would be welcomed.
- Can the authors say something about the variance in persuasiveness of the 50 messages in each of their AI/human-generated conditions in each of studies 1 and 2? I know they will be underpowered to precisely estimate their individual effects, but perhaps a histogram of the mean outcome of each message could be put in the appendix? For example, I'm picturing, for studies 1 and 2, one histogram for the human-generated messages and one histogram for the AI-generated messages — overlaid on the same plot. That way we could probably see just by eyeballing whether the variance is likely to be larger in the human or AI case.

Reviewer #3

(Remarks to the Author)

This review is for the manuscript "Artificial Intelligence Can Persuade Humans on Political Issues" (NCOMMS-23-33811). Overall, there are a lot of things to like about this manuscript: the research is incredibly timely, the manuscript itself is well-written, and the results are clear and compelling.

In general, I genuinely do not have much feedback for the authors. My intuition is that this manuscript is likely publishable at this journal as-is, which is something that I rarely (if ever) say. With that, then, please take the following as merely something to consider as a suggestion, but not as a prescription (or even necessarily a firm recommendation).

The primary area where I think a stronger contribution could be made to the literature would be to say a bit more about the "mechanisms" of persuasion in this context. That is: my sense is that there is a bit of room to go further into the domain of *why* the texts generated by the LLMs are more persuasive across these various issues. In its simplest form, I think that a slightly more rich interpretation of the LIWC results might be beneficial here — that is, what types of cues (or more categorically, "verbal behaviors") seem to be driving the effects that we see here.

One thing that I found particularly striking was how well-aligned the results are with other literature on verbal behavior and persuasion. For example, in past work with my colleagues, we find that more persuasive messages written by humans tend to have the following features:

1. More analytic / formal / logical
2. Higher readability scores
3. Longer texts
4. Less anecdotal (i.e., less self-referential language)

Ta, V. P., Boyd, R. L., Seraj, S., Keller, A., Griffith, C., Loggarakis, A., & Medema, L. (2022). An inclusive, real-world investigation of persuasion in language and verbal behavior. *Journal of Computational Social Science*, 5(1), 883–903. <https://doi.org/10.1007/s42001-021-00153-5>

Note that I am by no means suggesting that the authors cite our work specifically — our paper was more of an exercise in synthesizing an "inclusive" set of linguistic cues from a rather broad literature that has (traditionally) looked at these types of

cues in isolation. That is to say, this paper might be a good starting point to connect more deeply with the literature on persuasive cues in text to provide a more rich interpretation of the findings, if they feel that it is beneficial to do so.

The main reason that I suggest this potential avenue for going more deeply into these issues in the discussion section is that it might allow the authors to speak a bit more directly about what we can *do* with these findings. My current read of the discussion section is that it is leaning in the direction of recommendations for regulatory bodies to consider (and enact some type of controls/legislation) drawing from the current findings, highlighting how these emerging technologies can be used for good/bad applications alike. However, if we are able to nail down some more specific details about *why* or *how* the LLM-generated texts are more effective, we might be able to make some additional recommendations at other levels of society — for example, should researchers investigate whether persuasive appeals that have X, Y, and Z features should be flagged for deeper fact-checking? Should findings with broad scientific consensus be packaged more persuasively to meet potentially misleading or mal-aligned messages on more equal footing? What might we recommend the average person on the street do when they encounter political messages that are inordinately persuasive? And so on.

I acknowledge that this suggestion may also lead the manuscript away from the main goals/thrust that the authors would like the paper to accomplish — again, please take this as merely a suggestion rather than recommendation or expectation.

Small note: In the Discussion section, I interpreted the findings as showing that LLMs used more self-referential language:

"Further, several of these divergences were in line with traditional stereotypes of differences between artificial and human intelligence, with the authors of LLM-generated messages perceived to be more logical, better informed, less likely to tell stories, and more likely to use pronouns, including "I.""

However, in the supplemental materials, it appears that the AI-generated texts used fewer pronouns by quite a wide margin (unless I'm misreading the table, which is possible).

Overall, then, I find this to be a very strong manuscript, and I hope that the authors find this feedback helpful!

Sincerely,
Ryan L. Boyd

Reviewer #4

(Remarks to the Author)

I read the paper titled "Artificial Intelligence Can Persuade Humans on Political Issues" with great interest. This is an important and timely topic, and I applaud the authors for their thoughtful empirical package. I believe that with a thorough revision, this paper could make its way toward publication at Nature Communications. Below, I list my comments by section.

Introduction

- The authors introduced two tensions at the beginning of the paper. First, they started with the idea that political persuasion via AI can be good (e.g., democratizing engagement) or bad (e.g., scaling political disinformation). The second tension considered if LLM messages might be effective at changing people's views or not. The first tension (good vs. bad) takes up a lot of space in the manuscript and it is less central to the paper than the first tension. The authors might get more purchase out of making the tension related to "efficacy of AI vs. human messaging" more central. This would, in turn, streamline and focus the argument.

- Relatedly, I think the authors would improve their argument immensely by more clearly stating why they believe, based on prior evidence, text generated by AI to be more or less persuasively effective than humans at attitude change. There are several reasons why AI might be more effective at persuasion than humans. To name just a few: (1) AI has presumably learned what effective attitude change and/or persuasion looks like (its data is essentially the Internet). (2) AI has the potential to ignore biases that plague human persuasion (e.g., reliance on a range of heuristics that might lead people astray). There are also several reasons why AI might be less effective than humans at persuasion (e.g., without tailoring and training, AI responses can often be too general and abstract). Altogether, setting up this argument is the most important part of the paper's frontend, and I would encourage the authors to go deeper into the effectiveness question instead of the good vs. bad argument that is currently taking up more space.

- My most substantive comment on the Introduction: I encourage the authors to really consider if the paper and its results are indeed related to persuasion. Persuasion — supported by theories like the Elaboration Likelihood Model and others, for example — has very clear assumptions and conditions that need to be met for influence to occur (e.g., goals of the persuader, the message recipient's motivation and their capacity to process a message, among many others). These ideas were not measured in the current study, which is fine. But therefore, what I think we have here instead is more general attitude change that is facilitated by linguistic and communicative patterns of AI (more than human linguistic and communicative patterns). I would encourage the authors to reframe their piece around attitude change and not persuasion. I don't believe they lose anything by doing so, and there's more to gain (e.g., this change is more theoretically accurate and reflective of the work done).

Method and Results

- I'm fine with the authors using GPT-3 and GPT-3.5 given that the studies were collected in Nov-Dec 2022 (nice foresight!).

However, it might be important to consider GPT-4. I'm not suggesting the authors conduct another study with this model (this would be a waste of resources that could be used elsewhere), but they might regenerate AI responses with GPT-4 across studies and run correlations between GPT-3 and GPT-4 responses, and GPT3.5 and GPT-4 responses, along various linguistic measures of interest (from LIWC). I would imagine the correlations would be quite high, which will work in the author's favor. Consider comparing word count (structure), emotion (content), and analytic thinking (style), which should cover dominant types of language categories. This analysis would show that the results are likely not LLM dependent and there is good linguistic correspondence across LLMs. A thorny reader might question the results if GPT-4 isn't used (it's the latest shiny object in the LLM space), and this would hopefully assuage the reader and put the issue to bed.

- It would be helpful to understand why the specific topics in each study were chosen (Study 1 = smoking ban, Study 2 = assault weapons ban, Study 3 = different policies). I am sure that they authors thought about this decision quite a bit, so some "behind the curtain" commentary on why these topics were chose would be helpful (e.g., is there work that states these are the most contentious or polarized topics in the US? — I assume they are and that there is this work from Pew etc.).

- Thank you for providing both frequentist and Bayesian statistics – I appreciated the analytical clarity and comprehensiveness of this work. Well done!

- Did the authors measure existing knowledge about each topic? Topical knowledge is key to persuasion (or attitude change). I imagine most people would know enough about a smoking ban and how they feel toward it to write a coherent, plausible narrative and try to persuade another person (Study 1). An AI should be able to do this effectively as well (e.g., it should have data on smoking and health risks). However, in Study 3, it's unclear to me if humans know whether the U.S. should have a federal carbon tax or not. Do they even know what this topic or issue is really about (climate change)? How could one write a persuasive narrative or change the attitudes of another person on the topic if they don't know about the issue? In Study 3 (and perhaps others), one could imagine that the AI-generated messages were more "persuasive" than humans because the AI knew about the topic and could present itself as being informed (e.g., it offered evidence). This contention is supported by the author perception analyses, where there was a marginally significant relationship between AI and "smart" ($p = .052$). Therefore, one possible mechanism here is the presence or absence of appearing knowledgeable/competent. AI should almost always present as knowledgeable because it's going to provide a stance to the reader and back it up with evidence (typically). Humans will be more varied in their responses and have limited knowledge about political issues, on average. Addressing the idea of existing knowledge within the context of these studies and results is important for the authors to consider.

- Moderation analyses in the supplement were very interesting and important. I would encourage the authors to incorporate them into the main text for some of the most key variables (party ID).

- The paper would greatly benefit from a figure/table in the main text that compares exemplar responses from GPT and humans across studies. This would help bring the paper to life, showing exactly how the linguistic responses differed. Right now, the reader is forced to do a lot of heavy lifting and reference a lengthy supplement and OSF to receive these answers. Also, what were the distributions of responses for Figures 1 and 2 across conditions? It would be helpful to plot the points in R as well, perhaps on a raincloud or violin plot.

- I enjoyed seeing the paper's preregistered analytic plan, which included controlling for pre-treatment attitudes as a fixed effect in each model. Are the results consistent if change scores are used as the DV? That is, if we use a formula like $POST - PRE$ to create a single change score per participant.

Discussion

- The small effect sizes were unsurprising to me as a psychology of language scholar. Most effects in the psychology of language field are small (e.g., Kern et al., 2014 in *Assessment*; Kramer et al., 2014 in *PNAS*; Markowitz, 2023 in *JPSP*). The authors mention this ("Although the effects of LLM-generated messages are small, they are comparable in magnitude to the effects of human-generated messages."), but for uninitiated scholars, I would encourage the authors to reference work stating how small effect sizes based on language effects are expected and normal. When we scale these effects to population-levels, that's where we get to impact (but person-level effects are quite negligible).

- Ultimately, I believe this paper is about the degree to which text generated by AI vs. text generated by humans can change attitudes about politically divisive topics. This is meaningfully different than what the title suggests and the current framing (which, in my opinion, is a bit misleading). I would encourage the authors to be clearer – it's the text from AI that is changing attitudes, not AI alone. Therefore, my concern with the title and some of the framing is that it's not precise about what is doing the persuading or attitude changing. If the title and framing is left as "AI can persuade..." this could imply that you had people interact with an AI and this changed their views.

- Like with most LLM papers I have come across, I am curious about the question of "why" these effects have emerged and what it means that certain language patterns are more associated with attitude change than others. Why might it be the case that the GPT models are behaving this way? What do the results tell us about the content of the model's training data? Some thoughtful speculation or evidence-based reasoning to address these big questions would go a long way in the General Discussion. I appreciated the LIWC analyses, which provide some perspective into this question, but I was left wanting more.

- What are we to do with the idea that text from AI can facilitate more attitude change than text from humans? Does this mean that we need to scale AI text detectors in political communication or other walks of life, to determine if the communicator is

human or not (e.g., are we more susceptible to persuasion or attitude change, now)? Or, is this a new normal and we need to just deal with the consequences? Practical significance is undervalued in work that has the potential for impact, and I would encourage the authors to provide some guidance as to what these results might mean for policy makers, political strategists, companies that allow the untethered posting of political messages (e.g., social media companies), and everyday citizens.

Altogether, I think this is a fine piece of scholarship. I hope that these comments are helpful and taken in the spirit of this reviewer being interested in and engaged with the work. The number of points raised should not be an indication of the number of fatal issues with the paper. Instead, please use the number of comments as a signal for the amount of thought the work has sparked and how I would like to see the paper reach the finish line.

Version 1:

Reviewer comments:

Reviewer #1

(Remarks to the Author)

I like it!

The authors really came through in this strong revision. The contribution of this manuscript over previous work is clear and well-articulated. There are additional analyses that provide more value. Importantly, the framing in the very title of the work (and in the text) is much more accurate. Well done!

I have some minor comments, but will be happy to see the manuscript in print whether or not the authors implement them. First, while there are major differences from the current work, I would like to bring the authors' awareness to work that did, indeed, show "that LLM-generated content can be used to change recipients' policy attitudes". This work was done by my own team in [1,2] and is using a completely different mechanism than the one used in this work. Nevertheless, our work shows that LLM-generated content can be used to change recipients' policy attitudes using different means. This manuscript's contribution is very distinct, but these earlier papers may warrant a mention here.

Second, Figures 1 and 2: I appreciate seeing the full scale but I personally find the accordion figures difficult to read and interpret. It is also hard to see differences between the conditions with the new figures. Maybe find a way that balances the original figures' clarity with the revision's updates.

Finally, I am sure someone will proof the footnotes but they are currently skipping some numbers (e.g. 26).

Congrats on a great revision -- I am very happy to see how this manuscript evolved.

[1] Maurice Jakesch, Advait Bhat, Daniel Buschek, Lior Zalmanson, and Mor Naaman. 2023. Co-Writing with Opinionated Language Models Affects Users' Views. In Proceedings of the 2023 CHI Conference on Human Factors in Computing Systems (CHI '23). <https://doi.org/10.1145/3544548.3581196>

[2] Sterling Williams-Ceci, Maurice Jakesch, Advait Bhat, Kowe Kadoma, Lior Zalmanson, and Mor Naaman. Bias in AI Autocomplete Suggestions Leads to Attitude Shift on Societal Issues. <https://osf.io/preprints/psyarxiv/mhjn6>

Mor Naaman

Reviewer #2

(Remarks to the Author)

I previously reviewed this paper and was positively inclined towards it in that first review. In this revision the authors have addressed my three questions/concerns, which were:

1. Inclusion of covariate balance checks to demonstrate randomization worked as intended;
2. Noting a key limitation of their results is that the human messages were not written by experts;
3. Including graphs of the distribution of message effects across the full 50 messages in each of studies 1 and 2.

I thank the authors for addressing these points. I am satisfied with the contribution of the paper and endorse publication.

Reviewer #3

(Remarks to the Author)

Overall, I am very satisfied with the thorough revisions made by the authors. My impression is that they have more than adequately addressed the critiques that I raised in the previous round of reviews, and I congratulate the authors on their very nice work.

Sincerely,
Ryan L. Boyd

Reviewer #4

(Remarks to the Author)

I applaud the authors for their thorough, thoughtful, and careful revision of this manuscript. The authors were incredibly attentive to the requests of this reviewer (I was Reviewer 4 in the original submission). I also reviewed how the authors responded to requests from other referees, and I was impressed with their responses as well. Therefore, I recommend publication in Nature Communications. This paper has been substantially improved since its initial submission and it is primed to make an important impact across disciplines. I look forward to citing this piece (and teaching it, as well). Well done!

Sincerely,

David M. Markowitz

Reviewer #1 (Remarks to the Author):

1. This manuscript provides a clear and straightforward experiment that shows that AI can write text that could be as persuasive as human-written text. Overall, it is a well-written manuscript with pretty robust results. I do have some minor concerns about the manuscript, detailed below, but I have some greater concerns about the novelty of the work, and would like the authors to produce a stronger novelty argument for publication in a top journal.

We thank the reviewer for their supportive remarks and their excellent feedback that helped us to strengthen the paper. Below we respond to the concerns of the reviewer.

2. Generally, as noted above, I was very excited about this paper: although the setup is very simple, the novelty pitch was potentially interesting. However, to me, the findings -- including the core findings ("first evidence that LLM generated messages influence humans' support for policy issues") -- do not offer enough over the earlier work, most relatedly, the paper "Working With AI to Persuade: Examining a Large Language Model's Ability to Generate Pro-Vaccination Messages" (<https://dl.acm.org/doi/abs/10.1145/3579592>; I will refer to it here as the "vaccines paper"; it is not cited in the manuscript; it was written by members of the same Stanford institute). Can messages created by AI persuade? Yes, we already know that from the vaccines paper. Can they persuade about *policy issues*? Well if you consider vaccination attitudes "policy issues" (as I think one should), I believe the novelty of this current paper over the vaccine paper is again somewhat diminished. The vaccine paper even uses similar methods, first to evaluate the messages created (granted, not along the same dimensions, and they did not evaluate the policy attitudes) and then to compare their persuasion to human-created ones (again, granted, in a slightly different setup).

Taken together, I would like to see a much stronger novelty argument for this manuscript over this and other recent "AI-text persuasion" work. I would need a strong argument to be convinced this paper is appropriate for this venue; but that would be up to the editors to decide.

We appreciate the reviewer's feedback that we were not sufficiently clear on how our paper makes a contribution to the literature that goes beyond this previously published paper.

We have now revised our paper to make clear the critical ways in which our paper makes novel and unique contributions: (1) by demonstrating that AI-generated messages can persuade humans to *change* or *shift* their attitudes (as opposed to AI-generated messages being rated positively), (2) by demonstrating this persuasion on policy issues (as opposed to personal behavioral intentions, such as whether one wishes to get vaccinated), and (3) by demonstrating effective persuasion robustly across multiple unique issues, thus establishing the general claim that AI-generated messages can persuade humans on policy issues.

Regarding the first contribution, in the revised manuscript we now highlight that our effects are persuasion effects as defined by the relevant literature (see example definitions of persuasion from this literature in the next paragraph). While the “vaccine paper” establishes that LLMs can generate messages that humans rate positively (in terms of ratings being relatively high on an absolute scale), we demonstrate that LLMs can generate messages that persuade humans to *shift* their views on policies. In other words, the outcome variable in the vaccine paper is *positive perceptions of the vaccine message* (as measured by agreement with statements such as, “The message gave a reason for getting a COVID vaccine that is important to me”), whereas the outcome variable in our paper is *change in attitudes toward policies* (for example, measuring participants’ change in their responses to questions including “If there was a referendum tomorrow about a total smoking ban in all public places, how likely is it that you would vote in favor of a smoking ban?” before and after reading the messages).

Note that this is not merely an academic distinction as prior persuasion research (e.g., O’Keefe, 2018; Pink et al. 2023) finds that finding that a message is rated or perceived relatively positively by a recipient does not imply that the message is capable of influencing or persuading the person on the issue at hand. Thus, it is possible that messages are rated positively by a recipient, but do not change the recipient’s views.

This is a significant distinction in the literature on persuasion, where persuasion is treated as the process by which a *change* in beliefs, attitudes, or behaviors occurs. For example, three widely cited reviews of persuasion research define persuasion as “changing persons’ mental states” (O’Keefe, 2018) or “attitude change” (Petty & Cacioppo, 1986; Cialdini, 2001). The outcome variable in the vaccine paper would not qualify as attitude change, and therefore persuasion, according to these definitions, whereas our outcome variable does. We now clarify this point on page 4 of the revised manuscript, where we say “Despite much speculation regarding future applications of AI to politics, no prior work has yet tested whether exposure to LLM-generated messages can shift people’s views on policy issues. [...] In more recent research, researchers have also begun to

investigate how people rate messages generated by LLMs for their perceived persuasiveness in advertisements (Matz, 2023) or for promoting vaccine adoption (Karinshak et al., 2023). However, no work we are aware of has shown that LLM-generated content can be used to *change* recipients' policy attitudes, a key criterion in persuasion research (Petty & Cacioppo, 2012). [...] prior research finds that positive views of messages and messages' ability to persuade are often positively associated, but that increasing positive views of messages does not always result in attitude change (O'Keefe, 2018; Pink et al. 2023). Thus, for our goal of testing the capacity to persuade using AI-generated messages, testing the former would be insufficient to conclusively demonstrate persuasion.”

Second, while the vaccine paper focuses on personal health behavioral intentions (as measured by items such as “The message gave a reason for getting a COVID vaccine that is important to me”), our paper focuses on policy issues. We view both of these types of outcomes as academically interesting and societally important, and as having non-redundant value. Persuasion on policy attitudes is of unique interest, being a major area of academic study in the fields of social psychology and political science, a multibillion-dollar global industry, and a critical path to a number of societally important outcomes, e.g., election outcomes, ballot initiative outcomes, and social movement participation. Additionally, by demonstrating persuasive effects on policy issues we support our point that this newly established capacity of AI creates an urgency to understand and potentially regulate AI's use and misuse in political discourse. Consistent with this, our paper was cited in OpenAI's GPT-4 Technical Report as an example of work establishing the capacity for recent generations of LLMs to be deployed in political disinformation and/or propaganda campaigns. AI-facilitated policy influence could - and we expect likely will - have a number of societal ramifications, for example, facilitating various forms of misinformation campaigns, but also potentially democratizing citizens' abilities to advocate effectively for their views. While COVID vaccine intentions are an important and politicized issue that is *relevant to* policy, our paper addresses this open question more directly and conclusively by demonstrating AI-generated message persuasion on attitudes across six policy topics. In all cases, our measures are unambiguously about participants' support for the policies. To clarify these points, we have added the following on page 3:

“In more recent research, researchers have also begun to investigate how people rate messages generated by LLMs for their perceived persuasiveness in advertisements (Matz et al., 2023) or for promoting vaccine adoption (As measured by agreement with statements such as, “The message gave a reason for getting a COVID vaccine that is important to me”; see Karinshak et al, 2023). However, no work we are aware of has shown that LLM-generated content can be

used to *change* recipients' policy attitudes, a key criterion in persuasion research (Petty & Cacioppo, 2012).

“These distinctions are not trivial, neither practically nor scientifically. Practically, the most widely discussed applications of AI-generated persuasive content – for example, AI-generated misinformation or political campaign materials – would involve messages intended to shift attitudes on political issues and topics. Testing existing LLMs' capability of generating messages that can shift attitudes is a more direct demonstration of these applications. Theoretically, prior research finds that positive views of messages and messages' ability to persuade are often positively associated, but that increasing positive views of messages does not always result in attitude change (O'Keefe, 2018; Pink et al., 2023). Thus, for our goal of testing the capacity to persuade using AI-generated messages, testing the former would be insufficient to conclusively demonstrate persuasion.

“The current project focuses on persuasion on support for policies. Establishing the capacity for LLM-generated messages to persuade people on policy issues, even polarized issues, is highly significant. Public opinion on support for policies is a phenomenon of unique interest not only as a major area of academic study in the fields of social psychology and political science, but also a multibillion-dollar global industry, and a critical path to a number of societally important outcomes, including election outcomes, ballot initiative outcomes, and social movement participation. As such, we examine for the first time (to our knowledge) whether LLM-generated political appeals can be used to persuade humans on policy issues in terms of attitude change (Petty & Cacioppo, 2012).”

Third, we demonstrate that LLM-generated messages can be used to persuade humans across multiple issues, establishing the robustness of our central finding. Compared to a single-issue topic, our results demonstrate a more general point about policy persuasion, and some of the issues we study (e.g., gun control) have been polarized for decades, and thus are difficult to find persuasive effects on. To clarify this contribution, we write on page 12 of the revised manuscript: “We find such effects across different policies, including several highly polarized policies (e.g., an assault weapon ban) and a less polarized policy (a public smoking ban), using diverse samples that are representative of the U.S. population across three benchmarks (gender, race/ethnicity, and age). Therefore, the result that exposure to LLM-generated messages can lead to a change in policy attitudes is likely robust and generalizable.”

3. A more compelling path for a more significant contribution in this work is to develop the investigation of the persuasion aspect. Recent papers, for example, looked at the capabilities of LLM-based *personalization* to persuade (see "Evaluating the persuasive influence of political microtargeting with large language models", <https://osf.io/wnt8b/>, and "The Potential of Generative AI for Personalized Persuasion at Scale", <https://psyarxiv.com/rn97c>). These papers are new, and not yet (to my knowledge) peer reviewed. I do not expect the manuscript to refer to them, although that would be nice, for completeness, unless there is a reason not to. I am citing these papers here as examples of the type of novelty required in this area to justify a high-profile publication -- again, at least in my opinion.

We thank the reviewer for the suggestion to further develop the investigation of the persuasion aspect as a significant contribution. We describe the revisions that we made to address this point in response to Reviewer 1, Point 2.

We agree with the reviewer that the two linked working papers are valuable contributions to this new area of research. Notably, while the Hackenburg and Margetts paper studies the capacity for messages to influence individuals' attitudes, the Matz et al. paper studies perceptions of how persuasive messages are. As we note in our response to Reviewer 1, Point 2 right above, the latter is typically not characterized as a measure of persuasion in this literature.

Nonetheless, both papers are insightful and valuable. They focus most of all on the capacity of LLMs to be leveraged to create micro-targeted messages with potentially greater persuasive effects than non-micro-targeted messages, a very plausible potential application of LLMs that would capitalize on their enormous processing capacities. Interestingly, the two papers find conflicting results on the capacity for LLM-generated micro-targeted messages to be more persuasive.

We cited the recommended papers as we agree with the reviewer that doing so helps readers of our paper to find more relevant work in this area, and they can enrich our discussion of the potential for LLMs to be used in micro-targeting campaigns (a topic we only briefly speculated on before). We now do so on page 13, where we wrote:

“The effects of their applications could be further amplified by LLMs’ ability to customize messages based on the characteristics of recipients targeted for persuasion. Recent work reveals that LLMs can be used to generate customized messages to micro-target different recipients, though the evidence on its persuasive effect is still mixed. For example, Matz and colleagues (2023) show

that messages custom-generated by LLMs for recipients with different traits and ideological beliefs are perceived to be more persuasive by perceivers with the target traits and beliefs than by perceivers without them. However, Hackenburg and Margetts (2023) suggest that messages custom-generated by LLMs are no more effective in increasing support for policies than a generic message that was generated with a sophisticated prompt but not customized for a specific target audience. These inconsistencies in the effects of micro-targeting using LLMs suggest that more research may be required to understand its potential.”

4. These papers went further than investigating the text-creation capabilities of LLMs which, at this point, are well recognized. As the manuscript also notes, people cannot distinguish human- and AI-written text: so why would AI-written text be less persuasive? The interesting questions, to me, lie in the additional capabilities of LLMs: to produce and test messages cheaply and at scale.

We thank the reviewer for pushing us on this important point. Per the reviewer’s comment, we added emphasis on the point that politically persuasive messages can now be created cheaply at a massive scale. We added in the abstract: “These results demonstrate that the latest developments in LLMs have made it possible to create politically persuasive messages cheaply at a massive scale.” We also added on page 13: “We expect that applications of LLM-supported politically persuasive messages will be numerous in the future, given that politically persuasive messages can now be created cheaply at a large scale.”

5. A more minor contribution of the paper is showing that the AI-created text was different along several characteristics from human-created text. But it wasn't shown HOW that difference may have contributed to persuasion -- which would be a more significant contribution -- and the differences between AI and human text had been explored in other papers, e.g. "Human heuristics for AI-generated language are flawed" (cited in the manuscript as footnote 22).

We agree with the reviewer that the question of how the differences between AI-created text and human-created text shape persuasion is both significant and interesting. To strengthen our contribution regarding how AI-created text differs from human-created text, and how those differences may contribute to the persuasiveness of AI-generated versus human-generated messages, we conducted several mediation analyses. These analyses explore how participants’ perceptions of the messages’s authors were associated with observed persuasion effects.

Note that the analyses below summarize the results of mediation analyses focused on the AI-generated and human-generated message conditions. We are unable to conduct informative mediation analyses using our control conditions. This is because the texts used in the control conditions were not related to the political issues at hand, so comparisons of participants in the control conditions' perceptions related to non-political messages with participants in the AI- or human conditions' perceptions of political messages are not meaningful. For example, if we found that the effect of AI versus control messages on gun control policy attitude change was mediated by participants' perceptions that the AI-generated messages (on gun control) were more "factual" than the control message (e.g., on the history of skiing), we would not be able to conclude that AI's use of factual language in discussing gun control drove superior persuasiveness on the topic of gun control because the control messages were not about gun control. We explain this reasoning on page 36 in the revised SI:

“We are unable to conduct mediation analyses of what drives the persuasive effect of AI messages relative to the neutral messages in the control condition. This is because participants' responses to the questions about the features of the neutral control messages - which are on topics such as skiing and neckties - are not comparable to participants' responses in the human or AI conditions.”

Below is a summary of our analyses regarding participants' different perceptions of the authors, and how those different perceptions were related to observed attitude change in the studies. This summary appears in the revised manuscript on page 11, and further detail regarding the models can be found on page 9 of the SI:

“To understand the mechanisms underlying the persuasive effects of human- and LLM-generated messages, we also investigated how the authors of the human- and LLM-generated messages were perceived differently. Although participants who read messages generated by LLMs assumed these messages were human-generated, participants perceived the authors of LLM-generated messages differently than the authors of human-generated messages. As **Fig 2.** shows, recipients of LLM-generated messages, compared to human-generated messages, rated the authors as better informed ($b = 3.51$, 95% CI = [1.24, 5.79], $p = 0.003$), more logical ($b = 3.38$, 95% CI = [1.24, 5.52], $p = 0.002$), less angry ($b = -7.30$, 95% CI = [-10.00, -4.60], $p < 0.001$), less unique ($b = -4.09$, 95% CI = [-6.94, -1.25], $p = 0.005$), and less inclined to use story-telling ($b = -9.61$, 95% CI = [-12.6, -6.59], $p < 0.001$). We found no significant differences in participants' ratings of authors in the LLM and Human conditions on several other dimensions:

assertive, smart, moral, warm, authentic, creative, and interesting ($ps > 0.05$ for all; see Table SI5 in SI).

“We further conducted mediation analyses and found some evidence that each of the five perceptions for which we found significant differences were associated with persuasion effects. Importantly, these results should be interpreted with caution, as statistical mediation analysis is inherently limited in its ability to identify causal mechanisms (Bullock et al., 2010). These analyses suggest that, while LLM and human messages cause similar levels of attitude change, they do so through at least somewhat distinct pathways. The attitude change caused by exposure to LLM-generated messages (relative to human-generated messages) may be driven in part by the perceptions that their authors use more facts and evidence (indirect effect = 0.38, 95% CI = [0.05, 0.72]), more logical reasoning (indirect effect = 0.36, 95% CI = [0.02, 0.72]), and a dispassionate voice (less "angry"; indirect effect = 0.28, 95% CI = [0.12, 0.47]), consistent with past findings that messages using more logic-related words tend to be more persuasive (Xiao, 2018; Ta et al., 2022). By contrast, the persuasiveness of human-generated messages (relative to LLM-generated messages) may be driven in part by the perceived uniqueness (indirect effect = -0.62, 95% CI = [-1.04, -0.25]) and originality (indirect effect = -0.73, 95% CI = [-1.18, -0.35]) of the messages' authors. ”

The detailed results of the mediation analyses are described in detail in the SI's Mediators of the Persuasion Effects section. There, we also report the results of a simultaneous mediation analysis, a common analysis when testing multiple mediators. It showed that the anger pathway is the only significant indirect effect ($b = 0.25$, 95% CI = [0.06, 0.49]). However, the lack of the remaining pathways' effects need not indicate that they play no role in explaining the persuasion effects because, in a mediation model with multiple parallel mediators, one mediator's significance (while others are not) could indicate either (a) the significant mediator being the “real” or primary mediator, or (b) the significant mediator being the last in a causal chain involving multiple mediators. Therefore, the lack of indirect effects in the pathways involving the messages' being informed, logical, unique, or using vivid story-telling may be a result of their effects being further mediated by the anger pathway. Therefore we present these results only in the SI. However, if the editor and/or reviewers would prefer we change this presentation by moving the simultaneous mediation analysis to the main text, we are happy to do so and will highlight the ambiguity in interpreting results.

On page 37 of the revised SI, we also report that,

“We also considered testing other variables as mediators, including the policy evaluation measures (e.g., the policy is a “smart idea”) and objective text features (Linguistic Inquiry and Word Count, or LIWC). However, as reported in the main text, none of the policy evaluation measures differ across the AI and human conditions, disqualifying them as potential mediators. For objective text features, due to the small number of messages and the brevity of these messages, we were not able to conduct mediation analyses with sufficient power.”

6. Finally, the manuscript highlights in the discussion that "optimal persuasive messaging... involve leveraging the strengths of artificial and human intelligence together." This is interesting but a little bit expected (I do not know if I would call it "intriguing"). A more direct comparison would look at human-selected AI messages compared to human-selected human messages. Actually, why not let AI pick the most promising messages? That would be interesting as a comparison too.

We agree with the reviewer that we should tone down the language here and we changed the wording of “intriguing” to “additionally.” The sentence in the revised manuscript (page 12) now states “Additionally, we also found that a Human-in-the-Loop condition - in which humans tried to identify the most persuasive LLM-generated message to send to a participant - also performed well, nearly significantly outperforming LLM-generated messages in the internal meta-analysis ($p = .059$). Future research could compare the effect of human-selected LLM messages to human-selected human messages.” Also, we appreciate the reviewer pointing out several possibilities that we did not think about in the early stage of the research, and we added a short discussion on page 12: “Future research could investigate whether LLM-selected human messages or LLM-selected LLM messages can achieve even a greater level of persuasion effect.”

To me these are the main issues of this manuscript, touching on its novelty. There are other minor questions and places that require clarification that I list below, which I hope can be addressed with more careful reporting, listed next.

7. - I am afraid the very title of the work could be a bit misleading. The manuscript does not show that "Artificial Intelligence Can Persuade Humans on Political Issues"; it shows that "Text created by AI" can do the same. The difference is that people were not aware that they are getting AI-written arguments as suggested in the title (and in some of the manuscript text). As evident from the analysis, AI was not even a consideration for the participants (97% thought the writer was human, not surprisingly). That

participants thought the text was from humans is totally fine – but that makes the title not aligned with the main investigation. There is some language like that throughout, which I encourage revisiting (in some cases it is more palatable than others).

We thank the reviewer for this important point and changed the title to “AI-Generated Messages Can Be Used to Persuade Humans on Policy Issues”. We also changed the language related to this point throughout the paper.

For example, we changed “we investigate whether [...] LLMs are capable of influencing humans’ political attitudes” to “we investigate whether [...] LLMs can be used to create messages capable of influencing humans’ political attitudes.”

8. - I wonder what can be said about the potential impact of demand effects here. "participants reported their support for a policy before and after reading the assigned message". The participants thus know what they are asked to provide, and understand that they are being nudged to change their view. Is that a concern? Do other persuasion papers control for that in any way -- and, was anything done in this research to control for it?

We thank the reviewer for pressing us on this point. In the paper, we added a discussion of demand effects on page 13:

“Finally, a potential alternative mechanism for how AI-generated (and human-generated) messages cause attitude change is through demand effects. If demand effects are the main mechanism, these effects are more likely to generalize to natural contexts that inherently signal the message source’s intent (e.g., internet campaign ads and direct mail). However, recent research suggests that demand effects do not usually occur in online survey experiments (Mummolo & Peterson 2019), suggesting that our findings are unlikely driven by demand effects alone.”

9. - Similarly, using a 101-point policy support scale(!) seems to exaggerate the effect size for the figures. Is using this scale a common practice in this kind of study? The manuscript (or SI) should justify the choice of the scale -- I do not think I saw any justification. I also do not believe the actual number of points on the scale was ever mentioned (in SI or in OSF, but I may have missed it). Could the score be anything from 0 to 100? Or did it have 4, 5, 7, 9, 11 points? (see <https://pubmed.ncbi.nlm.nih.gov/10769936/>). Relatedly: it would also be useful to see histograms for the key DVs in the SI.

We agree with the reviewer that clarity on measurement is crucial. Per the reviewer's comment, we included the following justification on page 6 of the revised SI: "We used a 101-point scale because we anticipated small treatment effects. Such granular measures allow for capturing more nuanced differences (Evangelidis, 2023) and previous research suggests that 101-point scales allow for detecting small treatment effects in survey experiments (Voelkel et al., 2023)." We also revised the manuscript to make sure the number of points is mentioned where the results are reported for the first time on page 6 ("[...] the effect sizes were consistently small, ranging from about 2 to 4 points on the 101-point policy support scales we used in the three experiments")

We added histograms for the key DVs in the revised SI (see the "Distribution of the Pre-treatment Level of Support for the Policies" section).

10. - Speaking of data and distributions: the manuscript reported that "the issues are polarized" but how do we know that? What was the distribution of preliminary opinions in the data?

We thank the reviewer for helping us realize that we never substantiated our assumption that the issues we studied were polarized issues. We added a justification of this claim to the revised manuscript: "To increase generalizability, we used a wide range of policy issues, some of which are highly polarized", for which we added the following footnote (footnote 30) "See SI's "Distribution of the Pre-treatment Level of Support for the Policies" section." In the SI, we added distribution plots that describe the distribution of the attitudes by participants' party affiliation and added a section "Distribution of the Pre-treatment Level of Support for the Policies" on page 54-55, in which we included the following discussion:

"In the studies, we intended to test the persuasion effects on policies that vary in their levels of polarization. To verify our assumption that the issue in Study 1 (a smoking ban) is not very polarized and that Studies 2 and 3 are more polarized, we reviewed the mean scores of the pre-treatment level of support for these policies by participants' party affiliations and inspected the distributions.

"In Study 1, the mean level of pre-treatment support is 62.9 for Democrats, 52.1 for Independents, and 54.7 for Republicans with an 8.2 point gap between Democrats and Republicans. However, in Study 2, the mean pre-treatment support levels were 73.8 for Democrats, 49.6 for Independents, and 33.9 for Republicans, indicating a larger gap of 39.9 points between Democrats and Republicans.

Similarly, in Study 3, when collapsed across all 4 issues, the mean pre-treatment support levels were 77.2 for Democrats, 61.4 for Independents, and 46.6 for Republicans, further illustrating a notable partisan gap, here at 30.6 points between Democrats and Republicans.”

11. - I am confused about the reported numbers in the text, what tests/models they correspond to, and where do the numbers (e.g. "b = 3.62") come from. A short clarification may be needed.

We thank the reviewer for pointing out that our previous way of reporting the results was confusing. Per the reviewer’s comment, we added clarifications and explained in which SI table the estimates can be found. For example, we now clarified that the referenced “b = 3.62” can be found in Table SIIa in SI: “In Study 1, participants supported a smoking ban significantly more if they were assigned to the AI condition than if they were assigned to the Control condition ($b = 3.62$, 95% CI = [1.92, 5.32], $p < 0.001$; see Table SIIa in SI).”

12. - Also confusing: "LLM-generated messages were evaluated as positively as human-generated messages" - I think this refers to the evaluation of the policies, not the messages.

We thank the reviewer for catching this. We have rephrased the previously confusing sentence to: “LLM-generated messages led participants to evaluate the policies as positively as human-generated messages did.” (page 8).

13. - The analysis of the author characteristics is under "Features of the Messages". Why? Also, was there a correction for performing multiple tests here?

We agree with the reviewer that the previous section title was confusing. Per the reviewer’s comment, we changed the section title to “Perceptions of Messages, Message Authors, and Possible Mediation,” under which we also added the mediation analyses.

In the revised manuscript, we added that we conducted a robustness check using a correction for conducting multiple tests. We added a footnote (footnote 43) on page 9: “We also conducted a robustness check using a Bonferroni–Holm correction for these twelve analyses. All results are robust, except that the difference between AI-generated messages and human-generated messages for "unique" became marginally significant ($p = .060$).”

14. - How participants thought AI was involved in the writing is an interesting question. The details of how this question was asked are a little hidden in the SI but I think the text should be clear that when "we asked participants about who wrote the messages" only one of the several options was AI (as a reader, I thought you asked the "AI or human" question as used in previous studies). Regardless, about the statement "LLMs can readily generate messages perceived to be human-generated at the same rate as actually human-generated messages" -- I know this is not the manuscript's main claim but it is well documented by now in many papers. The point made in the discussion ("guessing that authors of both human-generated and LLM-generated messages were human at almost identical rates (94.7% and 94.4%, respectively) is therefore less interesting: the participants had no reason to believe AI was involved.

We thank the reviewer for helping us to put this finding in the context of the relevant literature. As such, we removed the sentence "This finding suggests that LLMs can readily generate messages perceived to be human-generated at the same rate as actually human-generated messages." and introduced the finding with the sentence "We also explored whether participants who read LLM-generated messages knew that they were reading LLM-generated messages in our research, as recent findings suggest that LLM-generated messages are often perceived to be human messages (Jakesch et al., 2023)" on page 9 of the revised manuscript.

Nonetheless, we believe that it is interesting for the reader to know these findings because we did find substantial differences between LLM- and human-generated messages (see our response to Reviewer 1, Comment 5).

Additionally, per the reviewer's comment, we now clarify on page 9 that "In Study 2, we asked participants about who most likely wrote the messages with the options of An adult person, A group of people, An expert on the topic, An artificial intelligence program, An intelligent adolescent, An elementary school-age child, and Other (please specific). Across conditions, the vast majority of participants believed that the messages were written by humans (i.e., options other than "An artificial intelligence program", LLM condition = 94.4%; Human condition = 94.7%; Human-in-the-Loop condition = 92.3%)"

15. - "LLM-generated messages used fewer pronouns, more long words" -- Why is that interesting? The manuscript does not set it up, or return to it in the discussion. (these results also align with previous work).

We agree with the reviewer that these results need to be better motivated. We have now expanded the discussion about the text analyses and provided more interpretations of the findings. It is now an independent section “Linguistic Features of the Messages” on page 12 of the revised SI:

“Additionally, we analyzed linguistic features of the LLM and human messages using dictionaries from Linguistic Inquiry and Word Count (LIWC) 2022 (Boyd et al., 2022). These analyses provide insights into if LLM and human-generated messages are linguistically distinguishable. In terms of structural language variables, LLMs used longer words (LLM_mean = 25.58% vs. Human_mean = 21.50% of words with more than 7 letters, $p < .001$, all p values corrected for multiple comparisons). LLMs and humans produced a comparable number of words per sentence (LLM_mean = 18.71 vs. Human_mean = 19.90, $p = .990$), but LLMs produced longer messages (LLM_mean = 237.08 vs. Human_mean = 200.07, $p < .001$). Using an open-source Python codebase (Schwartz et al., 2017), we extracted the Flesch-Kincaid reading level for the messages from both LLM and humans and found LLMs to use higher grade-level vocabulary (LLM_mean = 12.26 vs. Human_mean = 11.13, $p = 0.028$). We also found LLMs to produce more language reflecting “Analytic Thinking” (as a metric of logical, formal reasoning; LLM_mean = 82.82 vs. Human_mean = 58.85, $p < .001$), and less “Authentic” language (perceived honesty, genuineness; LLM_mean = 7.48 vs. Human_mean = 29.96, $p < .001$). This dovetails with the finding that the LLM uses fewer first-person singular (“I”) pronouns than humans (LLM_mean = 0.01 vs. human_mean = 0.66, $p < 0.001$), suggesting that humans rely more on appeals from personal experience. On the other hand, LLMs are more likely to use third-person singular (“we”) language (LLM_mean = 2.57 vs. 1.47 $p = 0.002$), likely in appeals to the common good. Interestingly, humans are more likely than LLMs to use third-person plural pronouns (“they”; LLM_mean = 1.00 vs. Human_mean = 1.83, $p < 0.001$), which is often used to refer to out-groups, and to use negations (human_mean = 1.93 vs. LLM_mean = 0.78, $p < 0.001$). In terms of overall “emotional tone,” however, we observed no significant difference between LLM and human language ($p > 0.900$). Overall, these results suggest that LLM-generated messages are on average more technical and analytical than human-generated messages and use a more sophisticated vocabulary. In contrast, human-generated messages rely more on personal appeals.”

16. Finally, the manuscript could use some streamlining and tightening. Two examples:

- In the abstract, the sentence "Our results show LLMs can persuade humans, even on highly polarized policy issues" -- seems repetitive with the previous text.

- The intro can also be streamlined. When the manuscript gets to "whether LLMs are capable of persuading people to shift their views on political issues" it's been already mentioned several times.

The sentence in the abstract is now revised as "These results demonstrate that the latest developments in LLMs have made it possible to create politically persuasive messages cheaply at a massive scale."

Additionally, we streamlined the manuscript so that "whether LLMs are capable of persuading people to shift their views on policy issues" is no longer mentioned several times."

Reviewer #2 (Remarks to the Author):**Summary**

This paper reports a series of three survey experiments with US adults in which the persuasiveness of political messages generated by the large language models GPT3 and GPT3.5 is tested against the persuasiveness of messages generated by humans (human condition), messages generated by AI and selected by humans (human-in-the-loop), or a control condition (placebo message). The different experiments study different political issues. Across the three experiments and corresponding political issues, the aggregate result is clear: the AI generated messages (1) persuade people on average and (2) are approximately as persuasive as the human-generated messages and the human-in-the-loop messages. In other words, the headline result is: yes, politically persuasive messages generated by GPT3 and 3.5 can exert a causal effect on US adults' political attitudes.

Comments

1. The contribution of this paper is clear and straightforward, and the headline result is convincing. Below I offer some observations on and suggestions that may improve the paper.

We thank the reviewer for their generous, supportive feedback as well as their suggestions for improving the paper.

2. I appreciated the balanced discussion of the possible positive/neutral/negative implications of AI-generated political communication throughout the paper. I have reviewed several related papers in recent months and too often it is assumed that the contribution of AI to political communication is necessarily going to be negative. This would not seem to be the case, and the current authors do a good job of offering a more balanced (and I think, accurate) perspective. They could perhaps go even further and explicitly say that persuasion is part and parcel of a healthy democratic politics — that is, people should not be concerned about persuasion per se. Rather, the concerns surround misinformation and misrepresentation embedded in the attempts at persuasion, as well as problems related to a small number of people “flooding the zone” with AI-generated content, as the authors note.

We thank the reviewer for this comment. We agree with the point that persuasion is an important part of functioning democracies. We revised our discussion of the role of

persuasion in a healthy democracy to highlight the reviewer’s points. On page 4, we now state “As persuasion is often part of healthy political communication, using LLMs could democratize political influence by helping people make more compelling appeals for their political views.”

3. I also appreciated the use of multiple issues across the experiments, which helps the generalizability of their estimates.

We thank the reviewer for this comment.

4. I think the authors should explicitly discuss an important limitation of their human-generated messages: they were written by laypeople, not political consultants or professional campaigners. This has direct implications for how AI may be taken up by political campaigns. For example, while AI may match laypeople for persuasiveness, it may still be outperformed by professionals. In this case, larger campaigns that have substantial resources may still favor hiring human professionals. And AI may disproportionately be used by smaller campaigns, who lack the resources to hire the comparatively more expensive professionals.

We agree with the reviewer that this is an important limitation which we now explicitly discuss in the revised manuscript. On page 12, we added that “A limitation of our studies is that the human-generated messages were written by laypeople, as opposed to political consultants or professional campaigners. Expert humans may produce more persuasive appeals than LLMs, at least for now. As a result, LLM-generated messages may be used disproportionately by campaigns with fewer resources. Future research is needed to compare the persuasiveness of more sophisticated LLMs to the persuasiveness of human experts.”

5. It would be nice to see covariate balance checks in the appendix for each study. The authors do a good job reporting post-treatment missingness in their outcome variables — and with the exception of study 1 it looks generally very low. Nevertheless, to reassure readers that randomization worked as intended, covariate balance checks would be welcomed.

Per the reviewer’s comments, we included a section, “Covariate Balance Checks”, in the SI (see below) that includes the covariate balance checks for each study. In short, we did not find any imbalance in Studies 1 and 3, though there is an imbalance in the pre-treatment level of support between the Human and the control conditions in Study 2 ($p = .010$). We consider this difference likely a result of random chance. More

importantly, this difference does not impact the interpretations of our results, as these pre-treatment differences were controlled for in the analyses specifically to address the possibility of randomization failure. These considerations are also included in the “Covariate Balance Checks” section in the SI:

“In Study 1, a one-way ANOVA revealed that the pre-treatment level of support does not differ by condition effect, $F(3, 1199) = 1.756, p = .154$.

“In Study 2, a one-way ANOVA revealed that the pre-treatment level of support differed by condition effect, $F(3, 2019) = 3.449, p = .016$. Post-hoc comparisons using the Tukey HSD test indicated that the mean score for the Human condition was significantly higher than that for the control condition (Mean difference = 6.34, $p = .010$). However, no other pairwise comparisons were statistically significant: control vs. LLM (Mean difference = -2.37, $p = .649$), Human-in-the-Loop vs. LLM (Mean difference = 1.49, $p = .885$), Human vs. LLM (Mean difference = 3.97, $p = .209$), Human-in-the-Loop vs. control (Mean difference = 3.86, $p = .226$), and Human vs. Human-in-the-Loop (Mean difference = 2.48, $p = .615$).

“In Study 3, four t-tests showed no significant difference in means between the control and LLM conditions for all four policies: $t(399) = -0.042, p = .966$ for the carbon tax, $t(383.44) = -0.633, p = .527$ for the paid parental-leave program, $t(399.98) = 0.522, p = .602$ for the child tax credit and $t(414.12) = 0.397, p = .691$ for the automatic voter registration policy.

“Though participants in the Human and control conditions in Study 2 differ in their pre-treatment level of support for the policy, we consider this difference likely a result of random chance. More importantly, this difference does not impact the interpretations of our results, as these pre-treatment differences were controlled for in the analyses specifically to address the possibility of randomization failure. Additionally, as shown above, in parallel models predicting the change in participants’ support for the policies (post-treatment support-pre-treatment support), the results are substantively the same, suggesting the robustness of our results.”

6. Can the authors say something about the variance in persuasiveness of the 50 messages in each of their AI/human-generated conditions in each of studies 1 and 2? I know they will be underpowered to precisely estimate their individual effects, but perhaps a histogram of the mean outcome of each message could be put in the appendix? For example, I’m picturing, for

studies 1 and 2, one histogram for the human-generated messages and one histogram for the AI-generated messages — overlaid on the same plot. That way we could probably see just by eyeballing whether the variance is likely to be larger in the human or AI case.

As suggested by the reviewer, we added the histogram plot (see below) in the SI’s “Distribution of the Messages’ Persuasiveness in Studies 1 and 2” section, and referenced it in the following footnote (footnote 38) on page 6 of the main text: “The distribution of the messages’ persuasiveness (the difference between participants’ post-treatment and pre-treatment policy support) across the human and AI conditions in Studies 1 and 2 can be found in SI”. In the section, we wrote:

“The histograms in Figure SI3 tabulate the distribution of the persuasiveness (the difference between participants’ post-treatment and pre-treatment policy support) of the messages in Studies 1 and 2. Though our studies are underpowered to provide precise estimates, overall, the variance of the persuasiveness of the messages appears to be somewhat greater among human-generated messages than LLM-generated messages.”

Figure SI3. Distribution of Persuasion Scores of the Messages.

Reviewer #3 (Remarks to the Author):

1. This review is for the manuscript "Artificial Intelligence Can Persuade Humans on Political Issues" (NCOMMS-23-33811). Overall, there are a lot of things to like about this manuscript: the research is incredibly timely, the manuscript itself is well-written, and the results are clear and compelling.

In general, I genuinely do not have much feedback for the authors. My intuition is that this manuscript is likely publishable at this journal as-is, which is something that I rarely (if ever) say. With that, then, please take the following as merely something to consider as a suggestion, but not as a prescription (or even necessarily a firm recommendation).

We thank the reviewer for their supportive comments and their helpful feedback for improving the paper.

2. The primary area where I think a stronger contribution could be made to the literature would be to say a bit more about the "mechanisms" of persuasion in this context. That is: my sense is that there is a bit of room to go further into the domain of *why* the texts generated by the LLMs are more persuasive across these various issues. In its simplest form, I think that a slightly more rich interpretation of the LIWC results might be beneficial here — that is, what types of cues (or more categorically, "verbal behaviors") seem to be driving the effects that we see here.

We thank the reviewer for helping us strengthen the contributions of our manuscript. This comment echoes another comment raised by Reviewer 1's Comment 5, which we addressed above. In short, we added additional mediation analyses to enrich the discussions about potential mechanisms. Below is a summary of our analyses regarding participants' different perceptions of the authors, and how those different perceptions were related to observed attitude change in the studies. This summary appears in the revised manuscript on page 9, and further detail regarding the models can be found on page 55 of the SI:

“To understand the mechanisms underlying the persuasive effects of human- and LLM-generated messages, we also investigated how the authors of the human- and LLM-generated messages were perceived differently. Although participants who read messages generated by LLMs assumed these messages were human-generated, participants perceived the authors of LLM-generated messages differently than the authors of human-generated messages. As **Fig 2.** shows,

recipients of LLM-generated messages, compared to human-generated messages, rated the authors as better informed ($b = 3.51$, 95% CI = [1.24, 5.79], $p = 0.003$), more logical ($b = 3.38$, 95% CI = [1.24, 5.52], $p = 0.002$), less angry ($b = -7.30$, 95% CI = [-10.00, -4.60], $p < 0.001$), less unique ($b = -4.09$, 95% CI = [-6.94, -1.25], $p = 0.005$), and less inclined to use story-telling ($b = -9.61$, 95% CI = [-12.6, -6.59], $p < 0.001$). We found no significant differences in participants' ratings of authors in the LLM and Human conditions on several other dimensions: assertive, smart, moral, warm, authentic, creative, and interesting ($ps > 0.05$ for all; see Table SI5 in SI).

“We further conducted mediation analyses and found some evidence that each of the five perceptions for which we found significant differences were associated with persuasion effects. Importantly, these results should be interpreted with caution, as statistical mediation analysis is inherently limited in its ability to identify causal mechanisms (Bullock et al., 2010). These analyses suggest that, while LLM and human messages cause similar levels of attitude change, they do so through at least somewhat distinct pathways. The attitude change caused by exposure to LLM-generated messages (relative to human-generated messages) may be driven in part by the perceptions that their authors use more facts and evidence (indirect effect = 0.38, 95% CI = [0.05, 0.72]), more logical reasoning (indirect effect = 0.36, 95% CI = [0.02, 0.72]), and a dispassionate voice (less "angry"; indirect effect = 0.28, 95% CI = [0.12, 0.47]), consistent with past findings that messages using more logic-related words tend to be more persuasive (Xiao, 2018; Ta et al., 2022). By contrast, the persuasiveness of human-generated messages (relative to LLM-generated messages) may be driven in part by the perceived uniqueness (indirect effect = -0.62, 95% CI = [-1.04, -0.25]) and originality (indirect effect = -0.73, 95% CI = [-1.18, -0.35]) of the messages' authors.”

The results of the mediation analyses are described in detail in the SI's Mediators of the Persuasion Effects section. We also conducted a simultaneous mediation analysis, a common analysis when testing multiple mediators. It showed that the anger pathway is the only significant indirect effect ($b=0.25$, CI=[0.06, 0.49]). We qualified this finding with the following discussion: “However, the lack of the remaining pathways' effects need not indicate that they play no role in explaining the persuasion effects. This is because, in a mediation model with multiple parallel mediators, one mediator's significance (while others are not) could indicate either (a) the significant mediator being the “real” or primary mediator, or (b) the significant mediator being the last in a causal chain involving multiple mediators. Therefore, the lack of indirect effects in the pathways involving the messages' being informed, logical, unique, or using vivid story-telling may

be a result of their effects being further mediated by the anger pathway.” Therefore we present these results only in the SI. To give insight on which of the mediators might be stronger causal mechanisms, we present the indirect effects in text from individual mediation analyses. However, if the editor and/or reviewers would prefer we change this presentation by moving the simultaneous mediation analysis to the main text, we are happy to do so and will highlight the ambiguity in interpreting results.

As mentioned in the SI’s Mediators of the Persuasion Effects section, we also clarify that “we are unable to conduct mediation analyses of what drives the persuasive effect of AI messages relative to the neutral messages in the control condition. This is because participants’ responses to the questions about the features of the neutral control messages - which are on topics such as skiing and neckties - are not comparable to participants’ responses in the human or AI conditions.”

Finally, as discussed in the same section, “We also considered testing other variables as mediators, including the policy evaluation measures (e.g., the policy is a “smart idea”) and objective text features (Linguistic Inquiry and Word Count, or LIWC). However, as reported in the main text, none of the policy evaluation measures differ across the AI and human conditions, disqualifying them as potential mediators. For objective text features, any potential mediation analyses would occur on the message-level as opposed to individual-level. Due to the small number of messages and the brevity of these messages, we were not able to conduct mediation analyses with sufficient power.”

Despite the lack of ability to conduct mediation analyses using LIWC variables, we conducted additional analyses on LIWC variables and expanded the discussion of them, which is now an independent section “Linguistic Features of the Messages”:

“Additionally, we analyzed linguistic features of the LLM and human messages using dictionaries from Linguistic Inquiry and Word Count (LIWC) 2022 (Boyd et al., 2022). These analyses provide insights into if LLM and human-generated messages are linguistically distinguishable. In terms of structural language variables, LLMs used longer words (LLM_mean = 25.58% vs. Human_mean = 21.50% of words with more than 7 letters, $p < .001$, all p values corrected for multiple comparisons). LLMs and humans produced a comparable number of words per sentence (LLM_mean = 18.71 vs. Human_mean = 19.90, $p = .990$), but LLMs produced longer messages (LLM_mean = 237.08 vs. Human_mean = 200.07, $p < .001$). Using an open-source Python codebase (Schwartz et al., 2017), we extracted the Flesch-Kincaid reading level for the messages from both LLM and humans and found LLMs to use higher grade-level vocabulary (LLM_mean = 12.26 vs. Human_mean = 11.13, $p = 0.028$). We also found LLMs to produce more

language reflecting “Analytic Thinking” (as a metric of logical, formal reasoning; LLM_mean = 82.82 vs. Human mean = 58.85, $p < .001$), and less “Authentic” language (perceived honesty, genuineness; LLM_mean = 7.48 vs. Human mean = 29.96, $p < .001$). This dovetails with the finding that the LLM uses fewer first-person singular (“I”) pronouns than humans (LLM_mean = 0.01 vs. human_mean = 0.66, $p < 0.001$), suggesting that humans rely more on appeals from personal experience. On the other hand, LLMs are more likely to use first-person plural language (“we”; LLM_mean = 2.57 vs. 1.47 $p = 0.002$), likely in appeals to the common good. Interestingly, humans are more likely than LLMs to use third-person plural pronouns (“they”; LLM_mean = 1.00 vs. Human_mean = 1.83, $p < 0.001$), which is often used to refer to out-groups, and to use negations (human_mean = 1.93 vs. LLM_mean = 0.78, $p < 0.001$). In terms of overall “emotional tone,” however, we observed no significant difference between LLM and human language ($p > 0.900$). Overall, these results suggest that LLM-generated messages are on average more technical and analytical than human-generated messages and use a more sophisticated vocabulary. In contrast, human-generated messages rely more on personal appeals.”

3. One thing that I found particularly striking was how well-aligned the results are with other literature on verbal behavior and persuasion. For example, in past work with my colleagues, we find that more persuasive messages written by humans tend to have the following features:

- 1. More analytic / formal / logical**
- 2. Higher readability scores**
- 3. Longer texts**
- 4. Less anecdotal (i.e., less self-referential language)**

Ta, V. P., Boyd, R. L., Seraj, S., Keller, A., Griffith, C., Loggarakis, A., & Medema, L. (2022). An inclusive, real-world investigation of persuasion in language and verbal behavior. *Journal of Computational Social Science*, 5(1), 883–903. <https://doi.org/10.1007/s42001-021-00153-5>

Note that I am by no means suggesting that the authors cite our work specifically — our paper was more of an exercise in synthesizing an “inclusive” set of linguistic cues from a rather broad literature that has (traditionally) looked at these types of cues in isolation. That is to say, this paper might be a good starting point to connect more deeply with the literature on persuasive cues in text to provide a more rich interpretation of the findings, if they feel that it is beneficial to do so.

We thank the reviewer for their helpful comment. In the revised manuscript, we discuss the connection between our results and the literature on verbal behavior and persuasion on page 9: “The attitude change caused by exposure to LLM-generated messages (relative to human-generated messages) may be driven in part by the perceptions that their authors use more facts and evidence (indirect effect = 0.38, 95% CI = [0.05, 0.72]), more logical reasoning (indirect effect = 0.36, 95% CI = [0.02, 0.72]), and a dispassionate voice (less "angry"; indirect effect = 0.28, 95% CI = [0.12, 0.47]), consistent with past findings that messages using more logic-related words tend to be more persuasive (Xiao, 2018; Ta et al., 2022)”

4. The main reason that I suggest this potential avenue for going more deeply into these issues in the discussion section is that it might allow the authors to speak a bit more directly about what we can *do* with these findings. My current read of the discussion section is that it is leaning in the direction of recommendations for regulatory bodies to consider (and enact some type of controls/legislation) drawing from the current findings, highlighting how these emerging technologies can be used for good/bad applications alike. However, if we are able to nail down some more specific details about *why* or *how* the LLM-generated texts are more effective, we might be able to make some additional recommendations at other levels of society — for example, should researchers investigate whether persuasive appeals that have X, Y, and Z features should be flagged for deeper fact-checking? Should findings with broad scientific consensus be packaged more persuasively to meet potentially misleading or mal-aligned messages on more equal footing? What might we recommend the average person on the street do when they encounter political messages that are inordinately persuasive? And so on.

I acknowledge that this suggestion may also lead the manuscript away from the main goals/thrust that the authors would like the paper to accomplish — again, please take this as merely a suggestion rather than recommendation or expectation.

We appreciate the insightful comments from the reviewer regarding the potential implications of our findings and their application at various societal levels. As suggested, we have expanded our Discussion section to address the why and how behind the effectiveness of LLM-generated texts and to provide actionable recommendations for different stakeholders.

In response to Reviewer 3, Comment 2, we delved into the mechanisms that might underlie the persuasive power of LLM-generated texts.

In terms of recommendations for stakeholders, we expanded the discussion section, and wrote on page 14:

“This demonstration of the potential use of LLMs in political persuasion presents regulators with new urgency as they consider what uses of LLMs to regulate, and how. At this stage, it is difficult to anticipate with confidence how public policy should respond to this new capacity of artificial intelligence, in part because it is unclear how this capacity will be used, and which uses will have negative effects on society. However, it might be wise to invest resources in closely tracking the application of AI to political persuasion and developing technology for identifying LLM-generated text content. Potential regulatory responses may consider policies such as mandatory disclosure of using LLM-generated content, embedding identifiers or digital watermarks in LLM-generated content, training AI models to detect LLM-generated content, and designing extensive guardrails on publicly available LLMs to refuse tasks, such as generating arguments in favor of unfounded claims. Additionally, digital literacy education programs may integrate AI literacy into curricula to foster critical engagement with persuasive content (Gao et al., 2022). Media companies may adopt robust fact-checking processes for highly persuasive or influential content, particularly those bearing hallmarks of AI generation. It is important for future research to explore the feasibility and efficacy of these and other potential solutions.”

5. Small note: In the Discussion section, I interpreted the findings as showing that LLMs used more self-referential language:

"Further, several of these divergences were in line with traditional stereotypes of differences between artificial and human intelligence, with the authors of LLM-generated messages perceived to be more logical, better informed, less likely to tell stories, and more likely to use pronouns, including "I."

However, in the supplemental materials, it appears that the AI-generated texts used fewer pronouns by quite a wide margin (unless I'm misreading the table, which is possible).

We appreciate that the reviewer caught this error. We have revised the text on page 10 to clarify this point: “We also found LLMs to produce more language reflecting “Analytic

Thinking” (as a metric of logical, formal reasoning; LLM_mean = 82.82 vs. Human mean = 58.85, $p < .001$), and less “Authentic” language (perceived honesty, genuineness; LLM_mean = 7.48 vs. Human mean = 29.96, $p < .001$). This dovetails with the finding that the LLM uses fewer first-person singular (“I”) pronouns than humans (LLM_mean = 0.01 vs. human_mean = 0.66, $p < 0.001$), suggesting that humans rely more on appeals from personal experience.”

6. Overall, then, I find this to be a very strong manuscript, and I hope that the authors find this feedback helpful!

We do and thank the reviewer very much for their feedback!

**Sincerely,
Ryan L. Boyd**

Reviewer #4 (Remarks to the Author):

1. I read the paper titled “Artificial Intelligence Can Persuade Humans on Political Issues” with great interest. This is an important and timely topic, and I applaud the authors for their thoughtful empirical package. I believe that with a thorough revision, this paper could make its way toward publication at Nature Communications. Below, I list my comments by section.

We are grateful for the excellent feedback from the reviewer who helped us to improve the paper.

Introduction

2. - The authors introduced two tensions at the beginning of the paper. First, they started with the idea that political persuasion via AI can be good (e.g., democratizing engagement) or bad (e.g., scaling political disinformation). The second tension considered if LLM messages might be effective at changing people’s views or not. The first tension (good vs. bad) takes up a lot of space in the manuscript and it is less central to the paper than the first tension. The authors might get more purchase out of making the tension related to “efficacy of AI vs. human messaging” more central. This would, in turn, streamline and focus the argument.

We thank the reviewer for this excellent analysis. We agree that the first tension is less directly related to the experimental manipulation and might therefore be confusing. We have shortened our references to the implications of AI-generated messages. Instead, we focus more on the second tension. Specifically, we revised the manuscript on page 2 to describe this second tension in more detail:

“We had competing expectations for whether LLM-generated messages can be persuasive, and also whether they may be as - or even more - persuasive than human-generated messages. Although LLMs have made breakthroughs in recent years, there are also reasons to doubt that exposure to LLM-generated messages would lead to a shift in political views. The best available experimental research on political persuasion generally finds that human-authored messages have small effect sizes (Coppock; 2023; Tppin et al., 2023), and the effects of persuasive efforts by political campaigns are typically small or null (Kalla & Broockman, 2018). Further, political persuasion is a complex activity, potentially drawing upon a range of skills, including perspective-taking, knowledge of the topic,

logical reasoning, clarity of expression, and knowledge of effective interpersonal influence techniques, with success ultimately in the hands of the person receiving the persuasive appeal. Persuasion is also uniquely challenging in highly polarized settings, such as the contemporary U.S., where many views are strongly held and difficult to influence (Druckman, 2022).

“However, it is also possible that LLM-generated messages are as persuasive, if not more than, human-generated messages: LLMs can be used to produce language with a proficiency that often surpasses average human abilities in generating complex, coherent, and topical text (Open AI, 2023). Additionally, while human-authored messages show small effect sizes in persuasion, LLMs may overcome these human limitations. For example, LLMs’ data-driven approach, derived from vast amounts of internet content, might enable them to dispassionately identify the strongest evidence and reasoning from their knowledge base and implement the most effective persuasion strategies more systematically than humans. Because of the competing expectations on whether LLM-generated messages can be persuasive at all, we did not specify an a priori hypothesis about whether messages generated by LLM might be more or less persuasive than those generated by humans.”

3. - Relatedly, I think the authors would improve their argument immensely by more clearly stating why they believe, based on prior evidence, text generated by AI to be more or less persuasively effective than humans at attitude change. There are several reasons why AI might be more effective at persuasion than humans. To name just a few: (1) AI has presumably learned what effective attitude change and/or persuasion looks like (its data is essentially the Internet). (2) AI has the potential to ignore biases that plague human persuasion (e.g., reliance on a range of heuristics that might lead people astray). There are also several reasons why AI might be less effective than humans at persuasion (e.g., without tailoring and training, AI responses can often be too general and abstract). Altogether, setting up this argument is the most important part of the paper’s frontend, and I would encourage the authors to go deeper into the effectiveness question instead of the good vs. bad argument that is currently taking up more space.

We appreciate the reviewer’s comment and expanded the discussion about the efficacy of AI vs. human messaging (see our response to Reviewer 4, Comment 2).

4. - My most substantive comment on the Introduction: I encourage the authors to really consider if the paper and its results are indeed related to persuasion. Persuasion — supported by theories like the Elaboration Likelihood Model and others, for example — has very clear assumptions and conditions that need to be met for influence to occur (e.g., goals of the persuader, the message recipient’s motivation and their capacity to process a message, among many others). These ideas were not measured in the current study, which is fine. But therefore, what I think we have here instead is more general attitude change that is facilitated by linguistic and communicative patterns of AI (more than human linguistic and communicative patterns). I would encourage the authors to reframe their piece around attitude change and not persuasion. I don’t believe they lose anything by doing so, and there’s more to gain (e.g., this change is more theoretically accurate and reflective of the work done).

We thank the reviewer for encouraging us to think more carefully about how our findings are related to persuasion. We reviewed the multidisciplinary literature on persuasion to understand how persuasion is defined and used. We found some definitions that correspond to our prior usage, in which persuasion is essentially equivalent to attitude change. For example, Petty and Cacioppo (2012) define it as “any change in attitudes that results from exposure to a communication.”

However, consistent with the reviewer's point, our review supports the reviewer’s point that many definitions involve additional features, most commonly, that an agent *intends* to persuade a target. For example: Druckman (2022) defines persuasion as “a successful intentional effort at influencing another’s mental state through communication in a circumstance in which the persuadee has some measure of freedom.” Similarly, DellaVigna & Gentzkow (2010) define “a persuasive communication to be a message provided by one agent (a sender) with at least a potential interest in changing the behavior of another agent (a receiver)”. Furthermore, Cacioppo et al. (2018) define persuasion as “the active attempt by an individual, group, or social entity (e.g., government, political party, business) to change a person’s beliefs, attitudes, or behaviors by conveying information, feelings, or reasoning.”

We did not mean to suggest that AI had such intention in the construction of our studies. Rather, we used AI as a tool and found that tool could in fact be used to persuade humans. However, the intention to do so was ours, not AI’s (perhaps one could argue that we gave AI the intention to create these messages, but we think this argument would be a

stretch). Thus, according to the most commonly used definitions in the literature, AI did not persuade. Instead, we used it as a tool to persuade. We have now revised our usage throughout to be consistent with this understanding, making the following changes, as what we investigated is whether LLMs can be used by individuals with an intention to persuade as a tool for persuasion:

-The title now reads “AI-Generated Messages Can Be Used to Persuade Humans on Policy Issues”

-Previously we used phrases such as “LLM (or LLM-generated messages) can persuade (or can be persuasive, or are capable of persuasion)” in many places in the manuscript. We have changed these to now be in the format of “exposure to LLM-generated messages can shift people’s views...” or “LLMs can be used to create messages capable of ...” throughout the manuscript. For example, we changed “we investigate whether [...] LLMs are capable of influencing humans’ political attitudes” to “we investigate whether [...] LLMs can be used to create messages capable of influencing humans’ political attitudes.”

Additionally, to clarify our outcome of interest, we now write on page 3 “As such, we examine for the first time (to our knowledge) whether LLM-generated political appeals can be used to persuade humans on policy issues in terms of attitude change (Petty & Cacioppo, 2012)”.

Method and Results

5. - I’m fine with the authors using GPT-3 and GPT-3.5 given that the studies were collected in Nov-Dec 2022 (nice foresight!). However, it might be important to consider GPT-4. I’m not suggesting the authors conduct another study with this model (this would be a waste of resources that could be used elsewhere), but they might regenerate AI responses with GPT-4 across studies and run correlations between GPT-3 and GPT-4 responses, and GPT3.5 and GPT-4 responses, along various linguistic measures of interest (from LIWC). I would imagine the correlations would be quite high, which will work in the author’s favor. Consider comparing word count (structure), emotion (content), and analytic thinking (style), which should cover dominant types of language categories. This analysis would show that the results are likely not LLM dependent and there is good linguistic correspondence across LLMs. A thorny reader might question the results if GPT-4 isn’t used (it’s the latest shiny object in the LLM space), and this would hopefully assuage the reader and put the issue to bed.

We thank the reviewer for this important point.

Per the reviewer’s suggestion, we conducted the additional analyses, and we added the following as a footnote (footnote 53) on page 12.

“Since the conclusion of our research, more powerful LLM models have been developed and become accessible to the public. We compared Study 1’s messages generated using GPT 3 on December 3, 2022, with messages generated using GPT 4 (“ChatGPT”) on November 8 2023 using the same prompt. We found there is a great deal of linguistic similarity between these LLMs (see the “Comparison of GPT 3 and GPT 4 Messages” section in SI). As such, it is likely that the effects we reported in the main text will persist if more recent models such as GPT 4 are used for generating messages.”

On page 45 of the revised SI, we added the referenced section “Comparison of GPT 3 and GPT 4 Messages”:

“Since the conclusion of our research, more powerful LLM models have been developed and become accessible to the public. For example, our research used GPT 3 and GPT 3.5 which were available at the time of our research in late 2022, whereas GPT 4 is widely available at the time of preparation of the below analyses of this document in late 2023. To probe into the question of whether our results are dependent on LLM models, we compared the 50 messages generated from Study 1 using GPT 3 on October 26, and 50 messages generated using GPT 4 (“ChatGPT”) on November 8 2023 using the same prompt. We found there is a great deal of linguistic similarity between these LLMs. As such, it is likely that the effects we reported in the main text would also be observed if messages generated by more recent models, such as GPT 4, were tested.

“Specifically, we compared their features in Linguistic Inquiry and Word Count (LIWC) 2022 and summarized the results in Table SIII. Most features are similar across the messages from the two models. For example, they use a similar number of words reflecting “Analytic Thinking” and first-person pronouns ($ps > .900$). However, some differences still emerged from the comparisons. For example, GPT 4, on average, uses words that are longer ($p < .001$), and has more words per message (GPT 3 mean = 188.94, GPT 4 mean = 253.88). Additionally, GPT 4 use higher grade-level vocabulary than GPT 3 (GPT 3 mean = 10.57, GPT 4 mean = 12.42, $p < .001$). All analyses were t-tests and all p values were corrected using Benjamini-Hochberg adjustment for multiple comparisons.”

Table SIII. Comparison of the Linguistic Features of GPT 3 and GPT4

	GPT 3.5	GPT 4	p
Analytic	87.40	87.01	0.973

	GPT 3.5	GPT 4	p
Clout	49.83	59.52	0.176
Authentic	9.12	18.38	0.000
Tone	32.41	47.53	0.110
WPS	18.58	19.31	0.973
BigWords	26.92	30.25	0.001
Dic	81.02	83.38	0.006
Linguistic	58.66	60.20	0.973
function.	47.32	47.38	0.973
pronoun	5.09	7.35	0.000
ppron	2.00	2.95	0.043
i	0.00	0.01	0.973
we	1.07	2.06	0.009
you	0.01	0.09	0.330
shehe	0.00	0.00	
they	0.57	0.55	0.973
ipron	3.09	4.40	0.000
det	13.42	15.65	0.000
article	7.39	9.92	0.000
number	1.75	0.74	0.000
prep	15.69	13.72	0.000
auxverb	8.77	6.54	0.000
adverb	2.94	3.95	0.013
conj	5.25	5.13	0.973
negate	0.73	1.61	0.000
verb	11.08	9.05	0.000
adj	9.15	10.70	0.000
quantity	6.03	4.00	0.000
Drives	5.16	6.51	0.028

	GPT 3.5	GPT 4	p
affiliation	1.95	3.31	0.002
achieve	2.01	1.26	0.002
power	1.95	2.31	0.973
Cognition	9.67	10.39	0.973
allnone	0.69	1.33	0.001
cogproc	8.75	8.99	0.973
insight	1.30	1.77	0.110
cause	2.62	1.74	0.000
discrep	2.85	1.82	0.007
tentat	0.37	0.43	0.973
certitude	0.06	0.42	0.000
differ	1.54	2.88	0.000
memory	0.00	0.02	0.973
Affect	4.65	5.46	0.054
tone_pos	2.54	3.61	0.002
tone_neg	2.09	1.85	0.973
emotion	0.67	0.99	0.973
emo_pos	0.22	0.58	0.013
emo_neg	0.25	0.26	0.973
emo_anx	0.03	0.03	0.973
emo_anger	0.03	0.07	0.973
emo_sad	0.09	0.03	0.973
swear	0.00	0.00	
Social	4.55	9.19	0.000
socbehav	1.43	2.86	0.000
prosocial	0.92	1.15	0.973
polite	0.01	0.11	0.163
conflict	0.20	0.29	0.973

	GPT 3.5	GPT 4	p
moral	0.03	0.19	0.076
comm	0.14	0.31	0.325
socref5	2.95	5.80	0.000
family	0.08	0.18	0.973
friend	0.00	0.00	
female	0.01	0.04	0.973
male	0.00	0.02	0.973
Culture	1.26	0.35	0.000
politic	1.24	0.35	0.000
ethnicity	0.02	0.00	0.973
tech	0.00	0.00	
Lifestyle	1.36	1.01	0.973
leisure	0.09	0.17	0.973
home	0.04	0.01	0.973
work	0.82	0.56	0.973
money	0.55	0.28	0.973
relig	0.00	0.02	0.973
Physical	7.20	7.18	0.973
health	4.46	5.54	0.144
illness	2.17	1.35	0.160
wellness	0.21	0.46	0.160
mental	0.00	0.03	0.973
substances	1.29	0.94	0.973
sexual	0.01	0.04	0.973
food	0.03	0.10	0.973
death	1.03	0.10	0.000
need	0.25	0.68	0.005
want	0.03	0.05	0.973

	GPT 3.5	GPT 4	p
acquire	0.09	0.25	0.447
lack	0.01	0.01	0.973
fulfill	0.01	0.02	0.973
fatigue	0.00	0.01	0.973
reward	0.31	0.29	0.973
risk	3.28	1.78	0.000
curiosity	0.12	0.17	0.973
allure	3.10	2.69	0.973
Perception	8.84	8.94	0.973
attention	0.00	0.21	0.000
motion	0.35	0.70	0.008
space	8.40	7.82	0.973
visual	0.05	0.20	0.111
auditory	0.05	0.05	0.973
feeling	0.01	0.02	0.973
time	0.94	1.48	0.007
focuspast	0.76	0.41	0.434
focuspresent	5.31	5.25	0.973
focusfuture	0.32	0.42	0.973
Conversation	0.00	0.01	0.973
netspeak	0.00	0.00	
assent	0.00	0.01	0.973
nonflu	0.00	0.00	
filler	0.00	0.00	
AllPunc	12.62	15.42	0.000
Period	5.54	5.28	0.973
Comma	5.48	7.04	0.000
QMark	0.00	0.02	0.973

	GPT 3.5	GPT 4	p
Exclam	0.00	0.00	
Apostro	0.29	1.29	0.000
OtherP	1.32	1.80	0.716

6. - It would be helpful to understand why the specific topics in each study were chosen (Study 1 = smoking ban, Study 2 = assault weapons ban, Study 3 = different policies). I am sure that they authors thought about this decision quite a bit, so some “behind the curtain” commentary on why these topics were chose would be helpful (e.g., is there work that states these are the most contentious or polarized topics in the US? — I assume they are and that there is this work from Pew etc.).

We agree with the reviewer that being transparent about these methodological choices is important. We added additional explanations about the rationale for selecting a diverse range of topics on page 14 of the revised manuscript:

“For Study 1, we chose a policy that we expected to be relatively favorable to demonstrating persuasion effects. We chose the policy of a smoking ban because this policy is not highly polarized, nor widely discussed. For Study 2, we sought to establish the robustness of the persuasion effects. We chose the policy of an assault weapon ban because this policy is a more polarized and widely discussed issue. For Study 3, we sought to more thoroughly establish the robustness of our findings. Therefore, we chose several, different policies that are highly polarized and widely discussed: a carbon tax, an increased child tax credit, a parental leave program, and automatic voter registration.”

7. - Thank you for providing both frequentist and Bayesian statistics – I appreciated the analytical clarity and comprehensiveness of this work. Well done!

We thank the reviewer for this note!

8. - Did the authors measure existing knowledge about each topic? Topical knowledge is key to persuasion (or attitude change). I imagine most people would know enough about a smoking ban and how they feel toward it to write a coherent, plausible narrative and try to persuade another person (Study 1). An AI should be able to do this effectively as well (e.g., it should have data on smoking and health risks). However, in Study 3, it’s unclear to me if humans know whether the U.S. should have a federal carbon tax or not. Do they even know what this topic or issue is really about (climate change)? How could one write a persuasive narrative or change the attitudes of another person on the topic if they don’t know about the issue? In Study 3 (and perhaps others), one could imagine that the AI-generated messages were more “persuasive” than humans because the AI knew about the topic

and could present itself as being informed (e.g., it offered evidence). This contention is supported by the author perception analyses, where there was a marginally significant relationship between AI and “smart” ($p = .052$). Therefore, one possible mechanism here is the presence or absence of appearing knowledgeable/competent. AI should almost always present as knowledgeable because it’s going to provide a stance to the reader and back it up with evidence (typically). Humans will be more varied in their responses and have limited knowledge about political issues, on average. Addressing the idea of existing knowledge within the context of these studies and results is important for the authors to consider.

We thank the reviewer for this important comment. We agree that knowledge is an important consideration for drafting persuasive content. We indeed considered that some participants may not fully know about the topics in Study 3, so in the study, we explained to the participants what each of the policies was before they provided their ratings. On page 5 of the revised manuscript, we added the following footnote (footnote 35) to clarify it “It was explained to participants what each of these policies were before they responded in the study.” Unfortunately, we did not have human-generated messages in Study 3. Therefore, our data does not allow us to compare the difference between the AI-generated messages and human-generated messages in this study.

However, we added a discussion of subject knowledge as a potential moderator for differences in effectiveness between AI- and human-generated messages to page 12 of the revised Discussion section: “In sum, several variables may determine whether AI-generated messages outperform human-generated messages or vice versa, including LLM’s training data, human authors’ expertise, recipients’ knowledge about the persuasion topics, and their perception of the messages’ human identity. Future research should investigate these potential moderators.”

One implication of this argument is that human experts, who are likely to be more knowledgeable than human laypeople, might be more effective message writers than AI (consistent with Reviewer 2’s comment 4). We added a discussion of this point on page 12: “A limitation of our studies is that the human-generated messages were written by laypeople, as opposed to political consultants or professional campaigners. Expert humans may produce more persuasive appeals than LLMs, at least for now. As a result, LLM-generated messages may be used disproportionately by campaigns with fewer resources. Future research is needed to compare the persuasiveness of more sophisticated LLMs to the persuasiveness of human experts.”

9. - Moderation analyses in the supplement were very interesting and important. I would encourage the authors to incorporate them into the main text for some of the most key variables (party ID).

Per the reviewer's comment, we added the following results of the moderation analyses in the main text as well on page 8:

“We also found some evidence in exploratory analyses that LLM-generated messages are more effective among Democrats and supporters of the policy issues. Using aggregated data from Studies 1 and 2, we found that partisan identity (measured as a 7-point variable, recoded to range from 0 = “Strong Democrat” to 1 = “Strong Republican”) moderated the treatment effects. The Partisan Identity \times LLM (vs control) condition ($b = -5.15$, 95% CI = $[-8.14, -2.16]$, $p = 0.001$) and Partisan Identity \times Human (vs control) condition ($b = -3.97$, 95% CI = $[-6.98, -0.96]$, $p = 0.010$) interaction effects were both negative and significant, suggesting that Republican participants were less persuaded by the LLM-generated and human-generated messages. The Human-in-the-Loop condition \times Party ID interaction effect was not significant ($b = -2.43$, 95% CI = $[-5.48, 0.63]$, $p = 0.120$). In other moderation analyses, we also found evidence that pre-treatment policy support moderated the effect of LLM-generated messages. The pre-treatment policy support \times LLM condition interaction effect was significant ($b = 0.05$, 95% CI = $[0.02, 0.08]$, $p = 0.001$), suggesting that participants with higher pre-treatment support were more persuaded by the LLM-generated messages. The pre-treatment policy support \times Human condition ($b = 0.00$, 95% CI = $[-0.03, 0.03]$, $p = 0.994$) and the pre-treatment policy support \times Human-in-the-Loop condition interaction effects were not significant ($b = 0.01$, 95% CI = $[-0.02, 0.04]$, $p = 0.473$). Thus, in line with prior research (Druckman et al., 2013), pre-existing identities and attitudes can play a crucial role in the effectiveness of persuasive messages.”

10. - The paper would greatly benefit from a figure/table in the main text that compares exemplar responses from GPT and humans across studies. This would help bring the paper to life, showing exactly how the linguistic responses differed. Right now, the reader is forced to do a lot of heavy lifting and reference a lengthy supplement and OSF to receive these answers. Also, what were the distributions of responses for Figures 1 and 2 across conditions? It would be helpful to plot the points in R as well, perhaps on a raincloud or violin plot.

We thank the reviewer for this great suggestion! We agree that a Table (Table 1; see below) in the main text with exemplar responses from GPT and human that illustrates the linguistic differences.

Table 1. Example Messages from AI and Human.

Study 1 Smoking ban	
AI Example 1 ...According to the Centers for Disease Control and Prevention (CDC), cigarette smoking kills more than 480,000 Americans each year, and exposure to secondhand smoke causes another 41,000 deaths. A total smoking ban in public places would help to reduce these numbers by preventing exposure to secondhand smoke and making it less convenient for people to smoke...	Human Example 1 I held my grandmother's hand as she left this world. Her death was caused by lung cancer, after smoking for over 40 years...Smoking is not a right. Smokers do not deserve special treatment and everyone has the right to breathe clean air...
AI Example 2 ...Total smoking bans in public places are the most effective way to protect non-smokers from secondhand smoke exposure. They are also supported by a majority of Americans. A recent poll found that 74% of Americans support banning smoking in all public places, including bars and restaurants...	Human Example 2 ...My mom has asthma and even a little smoke can set off an attack. I think we should be mindful of others and not do things that can hinder their health. A total smoking ban would be good for everyone, and people would be healthier too. We've already outlawed it in bars and restaurants and buildings, why not take it that step further so we can eradicate smoking entirely and get us back on track health wise?
Study 2 Assault weapon ban	
AI Example 1 ...Furthermore, an assault weapon ban is supported by a majority of the public. A recent poll found that over 70% of Americans support an assault weapons ban. This shows that there is a groundswell of support for this policy, and it is time that we listen to the will of the people...	Human Example 1 The local funeral homes are booked for the next week. More small caskets are having to be ordered. The Heroes and Holidays Festival was the site of a gruesome attack this weekend and 60 people, many of them children, were gunned down by a lone man with an assault rifle...Only a total ban on assault weapons will stop the continuing massacres...
AI Example 2 ...Assault weapons are designed for one purpose: to kill as many people as possible as quickly as possible. They are the weapons of choice for mass shooters and have no place in our communities. These powerful weapons have caused untold damage and heartache, and we must take steps to prevent them from causing any more...	Human Example 2 ...I grew up in a hunting family... agreeing with the Second Amendment of the Constitution... These weapons are designed to kill and kill they do, horribly. School children are literally being blown apart, some only able to be identified by their shoes... We the citizens need to come together for them and for the sake of our society this madness needs to end now.
AI Example 3 ...Assault weapons are ... made for mass destruction and have been used in some of the worst tragedies our nation has ever seen...The assault weapon ban will help to reduce the number of weapons out in the public, making it more difficult for criminals to get their hands on them...Furthermore, the assault weapon ban will help to reduce the financial cost associated with gun violence. The cost of medical treatment, counseling, and law enforcement efforts associated with these weapons is incredibly high...	Human Example 3 Sandyhook, Colombine, Parkland, Uvalde and so many more. Shootings in grocery stores, nightclubs, concerts, and Fourth of July parades. When will enough be enough? This could all be avoided if we had an assault weapon ban... There was a mass shooting in New Zealand in 2019 and it shock the country. They immediately implemented gun laws...there hasn't been a mass shooting since...

To show the distribution of responses for Figures 1 and 2 across conditions, we have replaced these figures with violin plots.

11. - I enjoyed seeing the paper's preregistered analytic plan, which included controlling for pre-treatment attitudes as a fixed effect in each model. Are the results consistent if change scores are used as the DV? That is, if we use a formula like POST – PRE to create a single change score per participant.

We agree that this is a good robustness check that many readers might find of interest. The results are substantively the same when we use the change scores instead. There was no case in which the estimates for the conditions change by more than 0.50, and none of the *p* values crossed the significance threshold. We added these analyses in the SI (Tables SI1b, SI2b, and SI3b; see below).

Table SI1b. Study 1 Results: Predicting Change in Support for a Smoking Ban by Condition

		b	S.E.	p	95% CI	η^2 (in %)	β
Model 1	(Intercept)	0.30 ns	0.62	0.623	[-0.90 1.51]		
	Human-in-the-Loop condition	5.04 ***	0.91	<.001	[3.27 6.82]	2.52	0.19
	AI condition	3.60 ***	0.87	<.001	[1.90 5.30]	1.42	0.14
	Human condition	3.37 ***	0.86	<.001	[1.68 5.06]	1.26	0.14
F(3,1199)=11.51, R-squared=0.03, adjusted R-squared=0.03							
Model 2	(Intercept)	3.67 ***	0.66	<.001	[2.37 4.97]		
	Human-in-the-Loop condition	1.68 †	0.99	0.090	[-0.26 3.62]	0.32	0.06
	AI condition	0.24 ns	0.94	0.801	[-1.62 2.09]	0.01	0.01
F(2,893)=1.63, R-squared=0, adjusted R-squared=0							
Model 3	(Intercept)	3.91 ***	0.65	<.001	[2.64 5.18]		
	Human-in-the-Loop condition	1.44 ns	0.96	0.133	[-0.44 3.32]	0.39	0.06
F(1,574)=2.27, R-squared=0, adjusted R-squared=0							

Note. Model 1's reference group is the Control condition. Model 2's reference group is the Human condition (the Control condition participants are excluded). Model 3's reference group is the AI Condition (the Control and Human condition participants are both excluded). S.E. refers to standard error of the estimates. ****p*<0.001, ***p*<.01, †*p*<.10.

Table SI2b. Study 2 Results: Predicting Change in Support for an Assault Weapon Ban by Condition

		b	S.E.	p	95% CI	η^2 (in %)	β
Model 1	(Intercept)	-0.01 ns	0.41	0.988	[-0.80 0.79]		
	Human-in-the-Loop condition	2.52 ***	0.58	<.001	[1.39 3.65]	0.94	0.12
	AI condition	1.92 ***	0.58	<.001	[0.79 3.05]	0.54	0.09
	Human condition	2.63 ***	0.57	<.001	[1.51 3.76]	1.03	0.12
F(3,2019)=9.02, R-squared=0.01, adjusted R-squared=0.01							
Model 2	(Intercept)	2.63 ***	0.44	<.001	[1.76 3.49]		
	Human-in-the-Loop condition	-0.11 ns	0.63	0.856	[-1.34 1.12]	0.00	-0.01
	AI condition	-0.71 ns	0.63	0.258	[-1.94 0.52]	0.08	-0.03
F(2,1507)=0.74, R-squared=0, adjusted R-squared=0							
Model 3	(Intercept)	1.92 ***	0.45	<.001	[1.03 2.80]		
	Human-in-the-Loop condition	0.60 ns	0.63	0.346	[-0.65 1.84]	0.09	0.03
F(1,1000)=0.89, R-squared=0, adjusted R-squared=0							

Note. Model 1's reference group is the Control condition. Model 2's reference group is the Human condition (the Control condition participants are excluded). Model 3's reference group is the AI Condition (the Control and Human condition participants are both excluded). S.E. refers to standard error of the estimates. ****p*<0.001, ***p*<.01, †*p*<.10.

Table SI3b. Study 3 Results: Predicting Change in Support for Policy by Condition

		b	S.E.	p	95% CI	η^2 (in %)	β
Overall model	(Intercept)	0.94 *	0.43	0.028	[0.10 1.78]		
	AI condition	2.88 ***	0.38	<.001	[2.14 3.63]	3.44	0.18
	Paid parental leave	-0.48 ns	0.54	0.373	[-1.55 0.58]	0.05	-0.03
	Child tax credit	0.17 ns	0.54	0.753	[-0.89 1.23]	0.01	0.01
	Automatic voter registration	-1.96 ***	0.53	<.001	[-3.01 -0.91]	0.83	-0.11
F(4,1605)=19.23, R-squared=0.05, adjusted R-squared=0.04							
Carbon tax model	(Intercept)	0.51 ns	0.5	0.313	[-0.48 1.50]		
	AI condition	3.74 ***	0.71	<.001	[2.34 5.14]	6.48	0.25
	F(1,399)=27.64, R-squared=0.06, adjusted R-squared=0.06						
Child tax credit model	(Intercept)	0.48 ns	0.58	0.409	[-0.66 1.62]		
	AI condition	4.14 ***	0.82	<.001	[2.53 5.74]	6.00	0.25
	F(1,400)=25.55, R-squared=0.06, adjusted R-squared=0.06						
Paid parental leave model	(Intercept)	0.71 ns	0.56	0.211	[-0.40 1.82]		
	AI condition	2.38 **	0.8	0.003	[0.81 3.94]	2.25	0.15
	F(1,388)=8.92, R-squared=0.02, adjusted R-squared=0.02						
Automatic voter registration model	(Intercept)	-0.23 ns	0.51	0.647	[-1.23 0.77]		
	AI condition	1.32 †	0.71	0.065	[-0.08 2.73]	0.82	0.09
	F(1,415)=3.42, R-squared=0.01, adjusted R-squared=0.01						

Note. S.E. refers to standard error of the estimates. *** $p < 0.001$, ** $p < .01$, † $p < .10$.

Discussion

12. - The small effect sizes were unsurprising to me as a psychology of language scholar. Most effects in the psychology of language field are small (e.g., Kern et al., 2014 in Assessment; Kramer et al., 2014 in PNAS; Markowitz, 2023 in JPSP). The authors mention this (“Although the effects of LLM-generated messages are small, they are comparable in magnitude to the effects of human-generated messages.”), but for uninitiated scholars, I would encourage the authors to reference work stating how small effect sizes based on language effects are expected and normal. When we scale these effects to population-levels, that’s where we get to impact (but person-level effects are quite negligible).

We thank the reviewer for the recommendations for additional references, and we incorporated them in the paper’s discussion of this point. On page 6, we now state “As is common in the political persuasion literature (Tappin et al., 2023; Kalla & Broockman, 2018; Druckman, 2022), as well as research in the psychology of language (Kramer et al., 2014; Markowitz, 2023), the effect sizes were consistently small, ranging from about 2 to 4 points on the 101-point policy support scales we used in the three experiments”.

13. - Ultimately, I believe this paper is about the degree to which text generated by AI vs. text generated by humans can change attitudes about politically divisive topics. This is meaningfully different than what the title

suggests and the current framing (which, in my opinion, is a bit misleading). I would encourage the authors to be clearer – it’s the text from AI that is changing attitudes, not AI alone. Therefore, my concern with the title and some of the framing is that it’s not precise about what is doing the persuading or attitude changing. If the title and framing is left as “AI can persuade...” this could imply that you had people interact with an AI and this changed their views.

We thank the reviewer for this important point. We changed the title to “AI-Generated Messages Can Be Used to Persuade Humans on Policy Issues” as well the language related to it throughout the paper. Additionally, all of the phrases in the format “LLM (or LLM-generated messages) can persuade (or can be persuasive, or are capable of persuasion)” are changed to be in the format of “exposure to LLM-generated messages can lead to attitude change” or “LLM can be used to create messages that are capable of ...”.

14. - Like with most LLM papers I have come across, I am curious about the question of “why” these effects have emerged and what it means that certain language patterns are more associated with attitude change than others. Why might it be the case that the GPT models are behaving this way? What do the results tell us about the content of the model's training data? Some thoughtful speculation or evidence-based reasoning to address these big questions would go a long way in the General Discussion. I appreciated the LIWC analyses, which provide some perspective into this question, but I was left wanting more.

We agree with the reviewer that this is an important question. We have now expanded the discussion about why AI-generated messages may rely on more evidence and reasoning (page 2 of the revised manuscript):

“However, it is also possible that LLM-generated messages are as persuasive, if not more than, human-generated messages: LLMs can be used to produce language with a proficiency that often surpasses average human abilities in generating complex, coherent, and topical text (Open AI, 2023). Additionally, while human-authored messages show small effect sizes in persuasion, LLMs may overcome these human limitations. For example, LLMs’ data-driven approach, derived from vast amounts of internet content, might enable them to dispassionately identify the strongest evidence and reasoning from their knowledge base and implement the most effective persuasion strategies more systematically than humans. Because of the competing expectations on whether

LLM-generated messages can be persuasive at all, we did not specify an a priori hypothesis about whether messages generated by LLM might be more or less persuasive than those generated by humans.”

In the result section, on page 9 we wrote:

“We further conducted mediation analyses and found some evidence that each of the five perceptions for which we found significant differences were associated with persuasion effects. Importantly, these results should be interpreted with caution, as statistical mediation analysis is inherently limited in its ability to identify causal mechanisms (Bullock et al., 2010). These analyses suggest that, while LLM and human messages cause similar levels of attitude change, they do so through at least somewhat distinct pathways. The attitude change caused by exposure to LLM-generated messages (relative to human-generated messages) may be driven in part by the perceptions that their authors use more facts and evidence (indirect effect = 0.38, 95% CI = [0.05, 0.72]), more logical reasoning (indirect effect = 0.36, 95% CI = [0.02, 0.72]), and a dispassionate voice (less "angry"; indirect effect = 0.28, 95% CI = [0.12, 0.47]), consistent with past findings that messages using more logic-related words tend to be more persuasive (Xiao, 2018; Ta et al., 2022). By contrast, the persuasiveness of human-generated messages (relative to LLM-generated messages) may be driven in part by the perceived uniqueness (indirect effect = -0.62, 95% CI = [-1.04, -0.25]) and originality (indirect effect = -0.73, 95% CI = [-1.18, -0.35]) of the messages' authors. ”

We also added additional text analyses and provided more interpretations of the findings (now as an independent section “Linguistic Features of the Messages” on page 10).

“Additionally, we analyzed linguistic features of the LLM and human messages using dictionaries from Linguistic Inquiry and Word Count (LIWC) 2022 (Boyd et al., 2022). These analyses provide insights into if LLM and human-generated messages are linguistically distinguishable. In terms of structural language variables, LLMs used longer words (LLM_mean = 25.58% vs. Human_mean = 21.50% of words with more than 7 letters, $p < .001$, all p values corrected for multiple comparisons). LLMs and humans produced a comparable number of words per sentence (LLM_mean = 18.71 vs. Human_mean = 19.90, $p = .990$), but LLMs produced longer messages (LLM_mean = 237.08 vs. Human_mean = 200.07, $p < .001$). Using an open-source Python codebase (Schwartz et al., 2017), we extracted the Flesch-Kincaid reading level for the messages from both LLM and humans and found LLMs to use higher grade-level vocabulary (LLM_mean =

12.26 vs. Human_mean = 11.13, $p = 0.028$. We also found LLMs to produce more language reflecting “Analytic Thinking” (as a metric of logical, formal reasoning; LLM_mean = 82.82 vs. Human mean = 58.85, $p < .001$), and less “Authentic” language (perceived honesty, genuineness; LLM_mean = 7.48 vs. Human mean = 29.96, $p < .001$). This dovetails with the finding that the LLM uses fewer first-person singular (“I”) pronouns than humans (LLM_mean = 0.01 vs. human_mean = 0.66, $p < 0.001$), suggesting that humans rely more on appeals from personal experience. On the other hand, LLMs are more likely to use third-person singular (“we”) language (LLM_mean = 2.57 vs. 1.47 $p = 0.002$), likely in appeals to the common good. Interestingly, humans are more likely than LLMs to use third-person plural pronouns (“they”; LLM_mean = 1.00 vs. Human_mean = 1.83, $p < 0.001$), which is often used to refer to out-groups, and to use negations (human_mean = 1.93 vs. LLM_mean = 0.78, $p < 0.001$). In terms of overall “emotional tone,” however, we observed no significant difference between LLM and human language ($p > 0.900$). Overall, these results suggest that LLM-generated messages are on average more technical and analytical than human-generated messages and use a more sophisticated vocabulary. In contrast, human-generated messages rely more on personal appeals.”

15. - What are we to do with the idea that text from AI can facilitate more attitude change than text from humans? Does this mean that we need to scale AI text detectors in political communication or other walks of life, to determine if the communicator is human or not (e.g., are we more susceptible to persuasion or attitude change, now)? Or, is this a new normal and we need to just deal with the consequences? Practical significance is undervalued in work that has the potential for impact, and I would encourage the authors to provide some guidance as to what these results might mean for policy makers, political strategists, companies that allow the untethered posting of political messages (e.g., social media companies), and everyday citizens.

We appreciate the reviewer pushing us on this important point.

First, this comment helped us to realize that we need to clarify in the revised manuscript that we find similar magnitude effects of AI-generated messages and human-generated messages on attitude change among participants. We now explicitly state this on page 12 of the manuscript: “Although the effects of LLM-generated messages are small, they are comparable in magnitude to the effects of human-generated messages.”

Second, we revised our discussion of policy implications, while avoiding making strong claims related to policy beyond what we directly studied in the manuscript (in line with the editor's recommendation). Specifically, we now wrote on page 14:

“This demonstration of the potential use of LLMs in political persuasion presents regulators with new urgency as they consider what uses of LLMs to regulate, and how. At this stage, it is difficult to anticipate with confidence how public policy should respond to this new capacity of artificial intelligence, in part because it is unclear how this capacity will be used, and which uses will have negative effects on society. However, it might be wise to invest resources in closely tracking the application of AI to political persuasion and developing technology for identifying LLM-generated text content. Potential regulatory responses may consider policies such as mandatory disclosure of using LLM-generated content, embedding identifiers or digital watermarks in LLM-generated content, training AI models to detect LLM-generated content, and designing extensive guardrails on publicly available LLMs to refuse tasks, such as generating arguments in favor of unfounded claims. Additionally, digital literacy education programs may integrate AI literacy into curricula to foster critical engagement with persuasive content (Gao et al., 2022). Media companies may adopt robust fact-checking processes for highly persuasive or influential content, particularly those bearing hallmarks of AI generation. It is important for future research to explore the feasibility and efficacy of these and other potential solutions.”

16. Altogether, I think this is a fine piece of scholarship. I hope that these comments are helpful and taken in the spirit of this reviewer being interested in and engaged with the work. The number of points raised should not be an indication of the number of fatal issues with the paper. Instead, please use the number of comments as a signal for the amount of thought the work has sparked and how I would like to see the paper reach the finish line.

We thank the Reviewer for taking the time to help us improve the manuscript. We have taken them in exactly this spirit and found them very helpful in identifying how to strengthen the paper!

References

- Cacioppo, J. T., Cacioppo, S., & Petty, R. E. (2018). The neuroscience of persuasion: A review with an emphasis on issues and opportunities. *Social Neuroscience*, 13(2), 129-172.
- DellaVigna, S., & Gentzkow, M. (2010). Persuasion: empirical evidence. *Annual Review of Economics*, 2(1), 643-669.
- Druckman, J. N. (2022). A framework for the study of persuasion. *Annual Review of Political Science*, 25, 65-88.
- O'Keefe, D. J. (2018). Message pretesting using assessments of expected or perceived persuasiveness: Evidence about diagnosticity of relative actual persuasiveness. *Journal of Communication*, 68(1), 120-142.
- Petty, R. E., & Cacioppo, J. T. (2012). *Communication and persuasion: Central and peripheral routes to attitude change*. Springer Science & Business Media.
- Pink, S. L., Stagnaro, M. N., Chu, J., Mernyk, J. S., Voelkel, J. G., & Willer, R. (2023). The effects of short messages encouraging prevention behaviors early in the COVID-19 pandemic. *PLoS One*, 18(4), e0284354.

We appreciate Reviewer 1's feedback and believe it has greatly improved our manuscript. Specifically:

- We appreciate the suggestions for the additional citations, which we have added to the manuscript.
- We appreciate the suggestion that it is not as easy to see the difference between the conditions in the new figure. To address this, we re-plotted the figure and added a gray horizontal line passing 0 so that it is easier to visually identify the conditions' effects relative to each other.
- We addressed the inconsistent citation/footnote numbering in our manuscript and have corrected for this.

Reviewers 2-4 did not make further requests or suggested modifications, but their encouraging and supportive feedback was much appreciated!